# Beyond Theorem Proving:
# Formulation, Framework and Benchmark for Formal Problem-Solving

**Qi Liu** [1 2]  **Xinhao Zheng** [1 2]  **Renqiu Xia** [1]  **Xingzhi Qi** [1]  **Qinxiang Cao** [1]  **Junchi Yan** [1 2]

## Abstract

Large language models (LLMs) have achieved remarkable progress in mathematical reasoning, yet persistently suffer from hallucinations and erroneous logic. While formal theorem proving (FTP) shows promise in process-level reliability, it is limited to *verification* (checking known propositions). This leaves constructive problem-solving (finding unknown terms that satisfy specific conditions) underexplored and disconnected from process-level verifiability. To bridge this gap, we introduce **FPS** (*Formal Problem-Solving*), a principled framework to encompass the end-to-end problem-solving process in Lean 4. In FPS, the answer is an unknown metavariable coupled with a proof obligation, forcing it to be mathematically derived and verified. We further present **D-FPS** (*Deductive FPS*), which structures solving into forward derivation and backward verification, aligning more closely with human reasoning steps. Three benchmarks of over 1,000 problems are constructed for evaluation: **FormalMath500**, **MiniF2F-Solving**, and **PutnamBench-Solving**. We further propose **RPE** (*Restricted Propositional Equivalence*), a symbolic metric that evaluates semantic correctness beyond brittle string matching. Extensive experiments with state-of-the-art provers reveal that solving is significantly harder than proving, highlighting the "alignment tax" required to transition from loose validity checking to constructive, human-aligned reasoning. Code and data are available at https://github.com/Purewhite2019/formal_problem_solving_main.

This work was supported by Scientific Research Innovation Capability Support Project for Young Faculty (U40) of the Ministry of Education of China (SRICSPYF-ZY2025019). [1] School of Computer Science & School of Artificial Intelligence, Shanghai Jiao Tong University [2]Shanghai Innovation Institute. Correspondence to: Junchi Yan <yanjunchi@sjtu.edu.cn>.

*Proceedings of the 43rd International Conference on Machine Learning*, Seoul, South Korea. PMLR 306, 2026. Copyright 2026 by the author(s).

## 1. Introduction

Large language models (LLMs) have achieved remarkable success in mathematical reasoning (DeepSeek-AI et al., 2025a; Jaech et al., 2024; Lightman et al., 2023; Snell et al., 2024). However, they fundamentally lack a rigorous mechanism to guarantee the correctness of intermediate reasoning steps (Zheng et al., 2024) and final answers (Li et al., 2024b). Critically, relying on flawed reasoning for training data can degrade model capacity (Allen-Zhu & Li, 2024) and induce incorrect learning dynamics (Ye et al., 2024).

Formal theorem proving (FTP) (Yang et al., 2025) enables process-verifiable mathematical proofs (Figure 1 (b)). Yet, current FTP research is largely confined to *proving known propositions*. This leaves a significant gap in mathematical practice, where the primary goal is often *solving for unknowns*: constructing an explicit object satisfying the given constraints as well as demonstrating its correctness. Existing neuro-symbolic pioneers (Zhou et al., 2024; Tsoukalas et al., 2024) alleviate this by cascading informal reasoning with FTP: an LLM solves the problem informally, fills the answer into a pre-formalized statement, and then generates a proof for verification (Figure 1 (b)). While this *post-hoc verification* provides outcome-level supervision, the solution[1] remains external and unverified (Lanham et al., 2023).

To bridge this gap, we introduce **FPS** (*Formal Problem-Solving*), a principled framework that unifies discovery and verification. Unlike FTP, where the goal is fixed, FPS formulates the solving process as the construction of a dependent pair: the answer term and its correctness proof. To further align formal solving with human-like deductive reasoning, we propose **D-FPS** (*Deductive FPS*) for *find-all* problems. D-FPS decouples the process into a *forward derivation* stage (explicitly deriving the answer from conditions) and a *backward verification* stage. We provide theoretical guarantees for soundness in FPS and for expressiveness, soundness, and completeness in D-FPS.

A consequent challenge is evaluation. Logically valid answers may misalign with human intuition, which may result in false positives. For example, "$\lim_{n \to \infty} \frac{1}{n}$" itself is a valid

---

[1]*Solution* and *answer* refer to the reasoning steps and final answer, respectively.

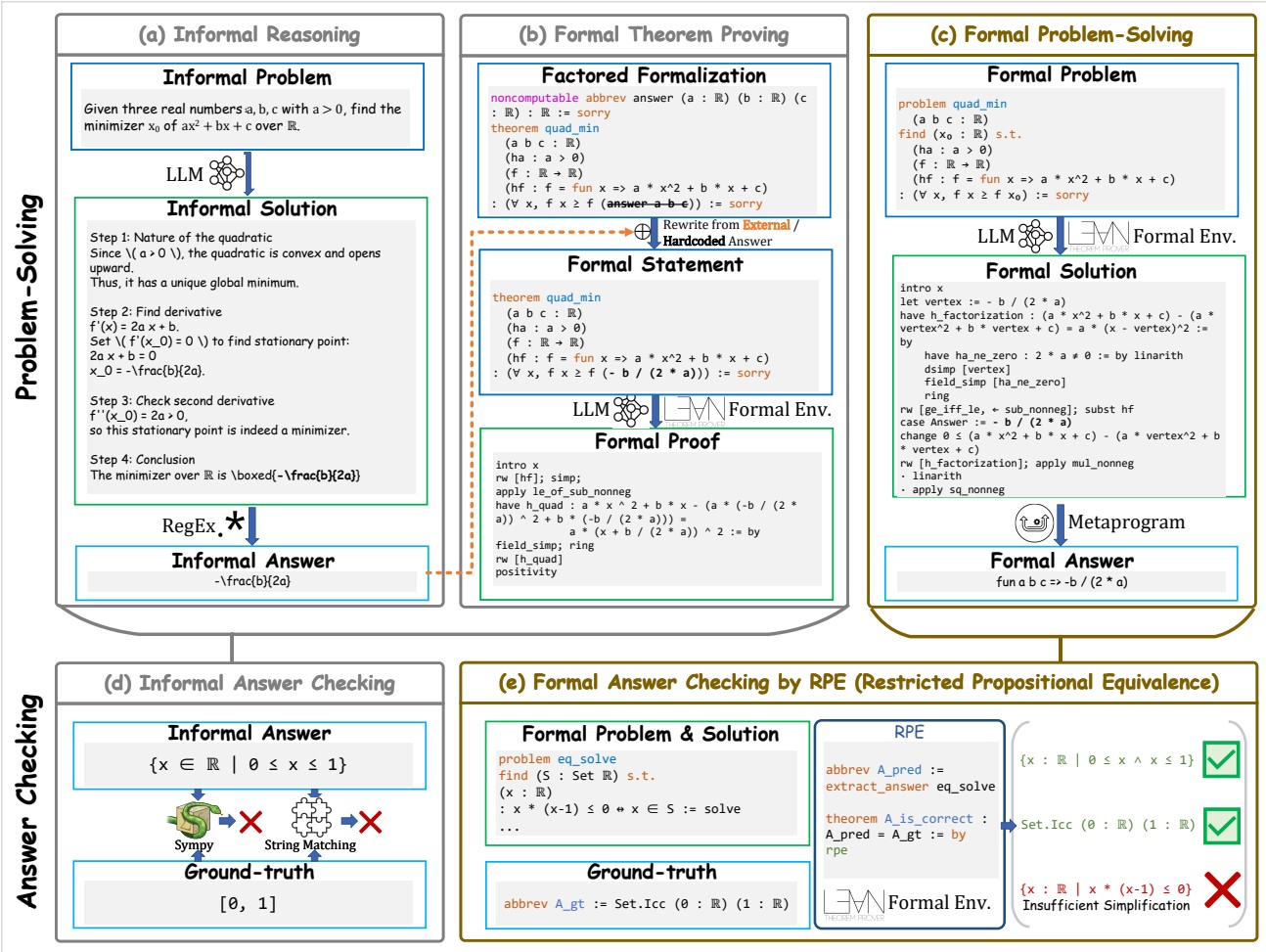

*Figure 1.* Overview of reasoning paradigms. **(a) Informal Reasoning** relies on LLM generation. While flexible, it suffers from hallucinations and relies on **(d)** brittle ad-hoc answer checking. **(b) Formal Theorem Proving** (FTP) is limited to verifying propositions but can be combined with informal reasoning for post-hoc verification. Using *factored formalization*, the answer is externally derived and hardcoded. The formal system only validates the result, leaving the derivation process unverified. **(c) Formal Problem-Solving** (FPS) (Ours) unifies derivation and verification by constructing both the answer term and its proof within the formal environment. **(e)** Restricted Propositional Equivalence (RPE) leverages the formal kernel to verify semantic correctness for arbitrary mathematical structures.

answer to "calculate $\lim_{n\to\infty} \frac{1}{n}$", but humans prefer "$0$". Existing informal answer checking methods, including exact match (Cobbe et al., 2021; Hendrycks et al., 2021; Li et al., 2024a), SymPy (Lewkowycz et al., 2022; Lightman et al., 2023; Glazer et al., 2024), and relaxed match (Gou et al., 2023), lack the expressiveness to handle complex mathematical objects, which may result in false negatives. To address these, we propose **RPE** (*Restricted Propositional Equivalence*), which leverages formal verification to determine semantic equivalence between two answer terms. Experiments validate RPE's high alignment with human judgment, with a Cohen's kappa of $0.9732$ (Cohen, 1960).

Furthermore, to facilitate research in this new domain, we address the data scarcity issue. Existing informal benchmarks focus on problem-solving and outcome-level correctness, while formal ones focus on theorem prov-

ing and process-level correctness. We propose a principled data annotation pipeline and construct three benchmarks with informal-formal parallel problems and answers: **FormalMath500**, a formalized subset of the MATH500 (Lightman et al., 2023) benchmark; **MiniF2F-Solving** and **PutnamBench-Solving**, refactored subsets of FTP benchmarks MiniF2F (Zheng et al., 2022) and PutnamBench (Tsoukalas et al., 2024).

Finally, we conduct extensive experiments benchmarking state-of-the-art FTP methods and prompting-based methods. Experiment results highlight that solving is significantly more challenging than proving. Moreover, we identify an "alignment tax": performance degrades when transitioning from loose proof construction in FTP to human-aligned answer construction in FPS.

## 2. Related Works

We defer additional related work on philosophy and answer checking to Appendix B for brevity.

**LLM for Informal Reasoning.** Recent works (Ouyang et al., 2022; Bubeck et al., 2023) show sparks of artificial general intelligence (AGI), but complex reasoning and problem-solving tasks (Huang et al., 2023; Xie et al., 2024a; Mirzadeh et al., 2024) remain challenging. Various methods are exploited to address this gap, including prompt engineering (Wei et al., 2022; Wang et al., 2022; Shinn et al., 2024), tool augmentation (Gou et al., 2023; Gao et al., 2023; Schick et al., 2023), and pre-/post-training (Azerbayev et al., 2023b; DeepSeek-AI et al., 2024; Luo et al., 2023). A recent trend is scaling not only at training but also inference (Jaech et al., 2024; Lambert et al., 2024), aided by verification engineering (Guan et al., 2024) and process supervision (Lightman et al., 2023; Wang et al., 2024; Lai et al., 2024; Zheng et al., 2024). However, (Stroebl et al., 2026) highlights that existing verifiers remain domain-limited or yield false positives. Therefore, we propose a neural-symbolic framework generalizable to well-defined problem-solving tasks, ensuring process-verifiable reasoning.

**LLM for Formal Reasoning.** Built on foundations like dependent type theory (Bove & Dybjer, 2008), formal theorem proving (FTP) environments such as Lean (Moura & Ullrich, 2021), Coq (Barras et al., 1997) and Isabelle (Nipkow et al., 2002) enable rigorous proof verification. Current FTP methods can be roughly divided into two paradigms. Proof search (Polu & Sutskever, 2020; Yang et al., 2024; Wu et al., 2025; Lample et al., 2022) constructs proofs step-by-step by transforming proof states. Whole-proof generation methods generate full proofs in one shot, either directly from formal statements (First et al., 2023; Xin et al., 2024a) or translated from informal proofs (Jiang et al., 2022; Wang et al., 2023; Zheng et al., 2023). FTP environments work as oracle verifiers (Stroebl et al., 2026) but are limited to proving known targets. Towards solving unknowns trustworthily, DTV (Zhou et al., 2024) and PutnamBench (Tsoukalas et al., 2024) adopt a *post-hoc verification* paradigm: an LLM first informally generates an answer and then attempts to prove its correctness. Recent works (Sivakumar et al., 2025; Sun et al., 2025) also study answer construction before proof. While these works make important progress, the reasoning process that produces the answer remains detached and unverified. In contrast, our work implements the end-to-end problem-solving process, including answer derivation, soundness/completeness proof, and correctness check, within formal verification.

**Symbolic Methods and Program-Aided Reasoning.** Symbolic systems like Mathematica (Kroll, 1989) and Z3 (de Moura & Björner, 2008), as well as neuro-symbolic solvers like AlphaGeometry (Trinh et al., 2024), are powerful but typically restricted to specific domains or lack human-readable deductive traces. Program-Aided Language models (PAL) (Gao et al., 2023; Gou et al., 2023) offload steps to Python interpreters. While effective for numerical calculation, they suffer from floating-point errors and cannot reason about abstract objects lacking computational realization. Our work overcomes these limitations by leveraging Lean 4 (Moura & Ullrich, 2021). Built on the expressive dependent type theory, Lean enables the formalization and verification of complex abstract structures, ensuring both rigor and generality.

**Benchmarks for Formal Reasoning.** The formal community has developed diverse benchmarks spanning various difficulty levels. MiniF2F (Zheng et al., 2022) and FIMO (Liu et al., 2023) target high-school to olympiad-level competitions, while ProofNet (Azerbayev et al., 2023a) and PutnamBench (Tsoukalas et al., 2024) focus on undergraduate mathematics. However, existing benchmarks focus on theorem proving or post-hoc verification (where an answer is informally generated and then verified), leaving the evaluation of process-level verified constructive solving (finding $a$ s.t. $P(a)$) underexplored. Our work bridges this gap by constructing specialized benchmarks, creating a dedicated testbed for end-to-end formal problem-solving.

## 3. Formalizing Problem-Solving

Preliminaries and background on Formal Theorem Proving (FTP) are deferred to Appendix A. In this section, we first present rigorous definitions of *problem* and *answer*, then introduce **FPS** (Formal Problem-Solving) to encompass the end-to-end problem-solving process in Lean 4, and finally propose **D-FPS** (Deductive FPS) to decouple derivation and verification.

### 3.1. Problem and Answer

Consider the following problem[2]:

**Example 3.1.** Given three real numbers $a, b, c$ with $a > 0$, find the minimizer $x_0$ of $ax^2 + bx + c$ over $\mathbb{R}$.

It consists of free variables $\{a, b, c\}$, a queried variable $\boxed{x_0}$, hypotheses $\{a > 0, f(x) = ax^2 + bx + c\}$, and conclusions $\{\forall x \in \mathbb{R}, f(x) \geq f(\boxed{x_0})\}$ that the $\boxed{x_0}$ must satisfy. The ground-truth answer $-\frac{b}{2a}$ is a term that depends on the free variables. All *elementary questions* (*whether-questions* and *which-questions*) (Belnap et al., 1976) can be expressed in this form (Nickles, 1981). Formally,

**Definition 3.2.** A ***problem*** $P(\hat{a}) = (\forall_{i=1}^{n} v_i, \bigwedge_{i=1}^{p} \phi_i \to \bigwedge_{i=1}^{q} \psi_i)[\boxed{a} \mapsto \hat{a}]$ is a predicate[3] that maps a direct answer

---

[2]More examples including yes-no question, equation, calculation, simplification, and counter-example are in Appendix D.1

[3]$\phi[x \mapsto t]$ represents substituting a term $t$ for a variable $x$ in a

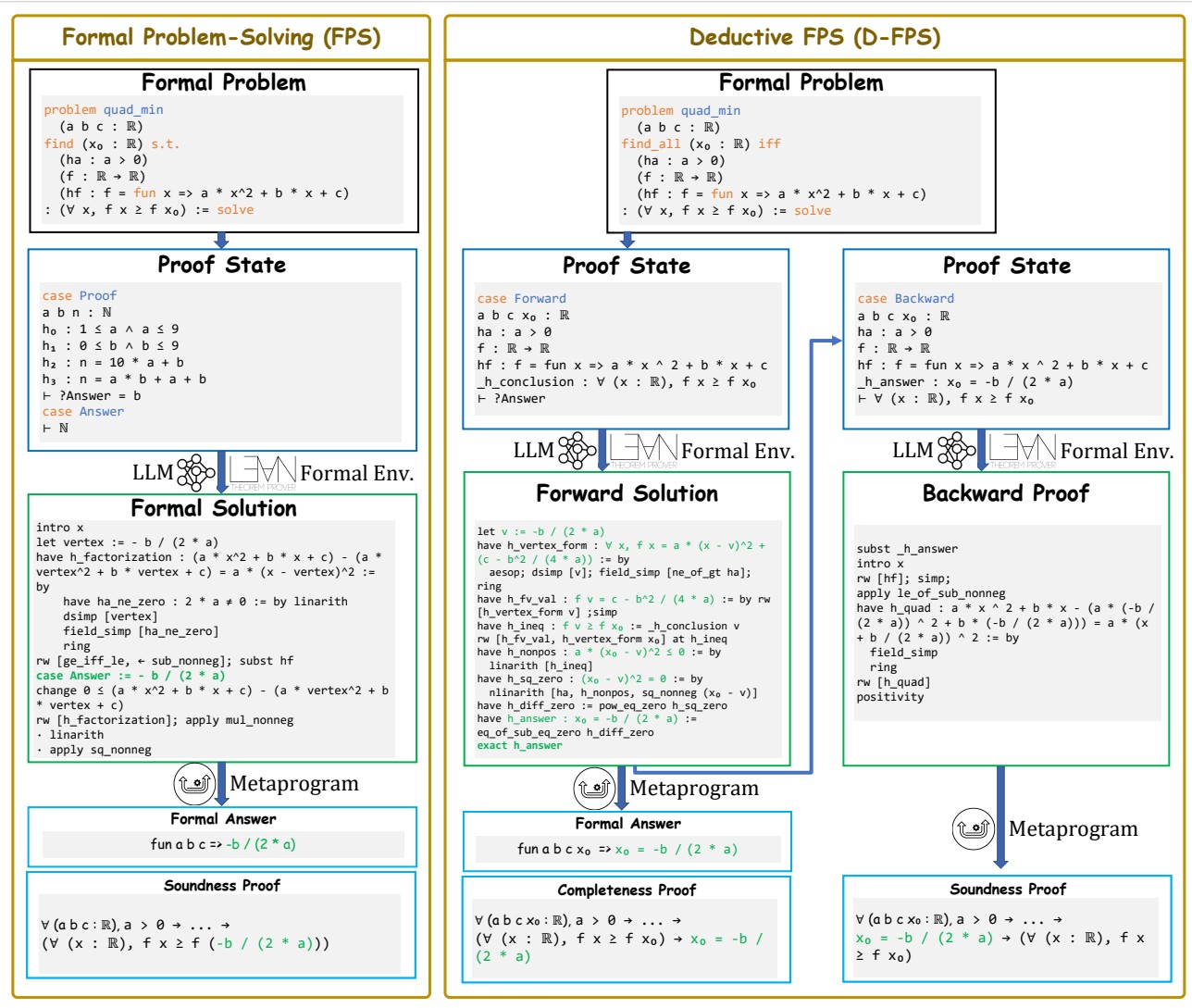

*Figure 2.* Workflow comparison between FPS and D-FPS. **(Left) FPS** treats solving as term construction. The proof state includes a metavariable `Answer` and a proof obligation `Proof`. `Answer` is instantiated either by direct assignment (potentially guessing) or implicitly via unification. **(Right) D-FPS** treats solving as logical derivation. The process is decoupled into a forward phase (deriving the candidate `Answer`) to ensure completeness (capturing all candidates), and a backward phase to ensure soundness (filtering spurious elements). Note that the instantiation of `Answer` is the end of the forward deduction, enforcing a rigorous derivation path.

$\hat{a}$ to a proposition. $P$ is composed of $(V, \boxed{a}, \Phi, \Psi)$, where

- *Independent variables* $V = \{v_i\}_{i=1}^n$ is the set of variables independent of $a$;
- *Queriable* $\boxed{a}$ is the queried variable that does not occur in $V$ and $\Phi$;
- *Hypotheses* $\Phi = \{\phi_i\}_{i=1}^p$ is the set of propositions that depend on $V$ (all of whose free variables are included in $V$), consisting of conditions that can be used to deduce the answer.
- *Conclusions* $\Psi = \{\psi_i\}_{i=1}^q$ is the set of propositions which depend on $V \cup \{\boxed{a}\}$, consisting of conclusions

---

formula $\phi$ at the places where $x$ occurs free.

that should be satisfied.

**Definition 3.3.** A *direct answer* $\hat{a}$ is a term that depends on $V$. Note that it is not necessarily a correct answer.

To avoid vacuous discussions of insolvable problems and following (Belnap et al., 1976), we presuppose the problems under discussion to be satisfiable.

**Assumption 3.4.** A problem is presupposed to be *satisfiable*, i.e., the following proposition holds.

$$\forall_{i=1}^n v_i, \bigwedge_{i=1}^p \phi_i \rightarrow \exists \boxed{a}, \bigwedge_{i=1}^q \psi_i \tag{1}$$

## 3.2. FPS: Formal Problem-Solving

While theorem proving focuses on verifying a proposition $P$, "problem-solving" is essentially delineating two requirements: **(1) Finding a direct answer** $\hat{a}$; **(2) Proving** $P(\hat{a})$. This corresponds to constructing an inhabitant of a $\Pi$-type wrapping a $\Sigma$-type (a dependent function returning a dependent pair) in Lean.

$$\Pi_{i=1}^{n}(v_i : T_i), \Sigma(\boxed{a} : T_a), \forall_{i=1}^{p}(h_i : \phi_i), (\wedge_{i=1}^{q}\psi_i) \quad (2)$$

where $T_i$ and $T_a$ are types of independent variable $v_i$ and queriable $\boxed{a}$, respectively. By Assumption 3.4, we assume the witness of $\phi_i$ to be $h_i$.

Therefore, Example 3.1 can be formalized as

$$\Pi(a\ b\ c : \mathbb{R}), \Sigma(\boxed{x_0} : \mathbb{R}), \forall(h_a : a > 0)(f : \mathbb{R} \to \mathbb{R})$$
$$(h_f : f = \lambda x.ax^2 + bx + c), (\forall x, f(x) \geq f(\boxed{x_0}))$$

We implement a domain-specific language (DSL) via metaprogramming to perform problem-solving in Lean:

```
problem ProblemName
  (v₁ : T₁) ... (vₙ : Tₙ)
find (⌐a⌐ : Tₐ) s.t.
  (h₁ : φ₁) ... (hₚ : φₚ)
: ∧�q_{i=1}ψᵢ := solve
```

The implementation of Example 3.1 is in Fig. 2 (upper left).

The DSL `problem ... find ... s.t. ...` expands into a definition of a $\Pi$-type wrapping a $\Sigma$-type as Eq. 2. The core tactic `solve` handles the initialization of the Lean 4 proof state by creating two subgoals:

- `Proof`: A proof goal where the agent proves $\forall_{i=1}^{n}(v_i : T_i), \forall_{i=1}^{p}(h_i : \phi_i), (\wedge_{i=1}^{q}\psi_i)$.
- `Answer`: A metavariable goal representing $\boxed{a}$ where the agent constructs an answer term based on $V$.

Naturally, a **solution** for the problem is a sequence of tactics that completes the proof state by solving the two subgoals, as demonstrated in Fig. 2 (middle left).

Upon successful problem-solving, the assignments to the `Answer` field and the `Proof` field are extracted via metaprograms `extract_answer` and `extract_proof`, respectively. The extracted answer and proof are functions that produce an answer and its soundness proof for each assignment of the independent variable $V$, as shown in Fig. 2 (bottom left).

**Theorem 3.5.** (Proof in Appendix E.1) *FPS is **sound**: for any problem $P$ and direct answer $\hat{a}$ produced by FPS, $P(\hat{a})$ holds.*

**Discussion.** Problems generally fall into two categories: (1) ***find-one*** (existential) problems, where any valid witness

suffices (e.g., constructing a counter-example); and (2) ***find-all*** (characterization) problems, which require identifying the complete set of solutions (e.g., finding all roots of an equation). FPS can naturally formalize *find-one* problems and guarantee *soundness*, i.e., any constructed witness $\hat{a}$ is valid. For *find-all* problems, our current implementation casts them as finding canonical objects that characterize the complete solution sets. However, direct FPS does not by itself guarantee *completeness* (all valid answers are covered).

Furthermore, the structural resemblance between FPS and standard FTP presents a trade-off. It allows existing FTP models to be applied directly without fine-tuning. However, since the `Answer` metavariable is coupled with the proof goal, unification-based automated tactics often implicitly instantiate the answer with tautological or incorrect terms just to close the proof, e.g., closing the proof goal "$\lim_{n\to\infty}\frac{1}{n} =?\boxed{a}$" by assigning $?\boxed{a} := \lim_{n\to\infty}\frac{1}{n}$. This implicit assignment diverges from the human preference for explicit deductive reasoning (Portoraro, 2025; Ahuja et al., 2025; Autexier & Dietrich, 2010), motivating the need for a more structured approach.

## 3.3. D-FPS: Deductive Formal Problem-Solving

To mitigate the limitations identified above (the risks of implicit assignment and incompleteness, and the divergence from human deductive reasoning), we introduce **D-FPS** (*Deductive FPS*). It shifts the paradigm from search-and-instantiate to derive-and-assign.

D-FPS is a structured FPS formulation that uses propositions instead of terms as answers, and decouples the solving process into two phases. Formally,

- The queriable $\boxed{A}$ lives in the `Prop` universe;
- $\Psi = \{\wedge_{i=1}^{q}\psi_i \leftrightarrow A\}$ and $A$ does not occur free in $\wedge_{i=1}^{q}\psi_i$;
- *Forward.* Fig. 2 (middle). The LLM must first explicitly derive a proposition $\boxed{A}$ from the hypotheses $\Phi$ and conclusions $\Psi$ to find the necessary conditions of the ground-truth proposition, i.e., capturing the full answer set (Completeness). The instantiation of $\boxed{A}$ is the end of the forward deduction, encouraging an explicit derivation path.
- *Backward.* Fig. 2 (right). The model verifies that the derived answer satisfies the conclusions $\Psi$, ensuring that it is a sufficient condition for the ground-truth proposition (soundness).

**Theorem 3.6** (Proof in Appendix E.2). *For find-all problems, D-FPS is at least as expressive as FPS.*

All find-all problems expressed in FPS can be converted into D-FPS while maintaining semantic equivalence, per Appendix E.2. As an illustration, Example 3.1 can be for-

malized as

$$\Pi(a\ b\ c\ x_0 : \mathbb{R}), \Sigma(A : \texttt{Prop}), \forall(h_a : a > 0)(f : \mathbb{R} \to \mathbb{R})$$
$$(h_f : f = \lambda x.ax^2 + bx + c), A \leftrightarrow (\forall x, f(x) \geq f(\boxed{x_0}))$$

We implement a DSL for D-FPS that shares a similar structure with FPS via metaprogramming:

```
problem ProblemName
    (v₁ : T₁) ... (vₙ : Tₙ)
find_all ( a  : Tₐ) iff
    (h₁ : φ₁) ... (hₚ : φₚ)
: ∧ᵠᵢ₌₁ψᵢ := solve
```

The implementation of Example 3.1 in D-FPS can be found in Fig. 2 (upper right).

The core tactic `solve` handles the initialization of the Lean 4 proof state by creating three subgoals:

- `Forward`: A proof goal where the agent proves $\forall_{i=1}^{n}(v_i : T_i), \forall_{i=1}^{p}(h_i : \phi_i)(h_c : \wedge_{i=1}^{q}\psi_i), \boxed{A}$.
- `Backward`: A proof goal where the agent proves $\forall_{i=1}^{n}(v_i : T_i), \forall_{i=1}^{p}(h_i : \phi_i)(h_a : \boxed{A}), \wedge_{i=1}^{q}\psi_i$.
- `Answer`: A metavariable goal representing $\boxed{A}$ where the agent constructs an answer term based on $V$.

Upon filling `Forward`, metaprograms can extract functions that map free variables to an answer proposition and its completeness proof. Upon proving `Backward`, the soundness proof can be further extracted, as shown in Fig. 2 (bottom).

**Theorem 3.7** (Proofs in Appendix E). *For any find-all D-FPS problem with ground-truth answer $\bar{A}$ and produced answer $\hat{A}$, D-FPS is:*

**complete:** $\forall_{i=1}^{n}(v_i : T_i), \forall_{i=1}^{p}(h_i : \phi_i), \bar{A} \to \hat{A}$,

*and, after backward verification,*

**sound:** $\forall_{i=1}^{n}(v_i : T_i), \forall_{i=1}^{p}(h_i : \phi_i), \hat{A} \to \bar{A}$.

**Intuition.** *Completeness* holds because the forward phase starts from the conclusions $\Psi$ and deduces the answer proposition $\hat{A}$. Since logic deduction preserves truth, any valid solution satisfying $\Psi$ must also satisfy $\hat{A}$. *Soundness* holds because the backward phase explicitly requires proving $\hat{A} \to \bigwedge_{i=1}^{q}\psi_i$, filtering out any spurious candidates introduced during the deduction.

### 3.4. RPE: Restricted Propositional Equivalence

Although FPS and D-FPS provide formal validity guarantees, validity alone does not guarantee utility. They can theoretically be "cheated" by the axiom of choice to extract a witness without computation, or by returning the problem statement itself as a tautology, e.g., answering "compute $\lim_{n\to\infty}\frac{1}{n}$" with "$\lim_{n\to\infty}\frac{1}{n}$ itself."

**Definition 3.8.** A direct answer $\hat{a}$ for problem $(V, \boxed{a}, \Phi, \Psi)$ is **correct** if: (1) $P(\hat{a})$ holds; (2) answering $(V, \boxed{a}, \Phi, \Psi)$ with $\hat{a}$ aligns with the expert-curated canonical answer.

To enforce alignment with human preference, we operationalize *correctness* as algorithmic closeness between a direct answer $\hat{a}$ and a human-annotated ground-truth $\bar{a}$. Since propositional equality (Moura & Ullrich, 2021) is so permissive that it allows the aforementioned tautologies, we propose **RPE** (*Restricted Propositional Equivalence*) to model correctness as provability within a **Restricted Tactic Set** $\mathcal{T}$. As illustrated in Fig. 1 (bottom), $\hat{a} \approx_{\text{RPE}} \bar{a}$ if and only if $\forall_{i=1}^{n}(v_i : T_i), \hat{a} = \bar{a}$ can be proven by $\mathcal{T}$, where $\mathcal{T}$ consists of canonical normalizers (`norm_num`, `ring_nf`), definitional checkers (`rfl`), and bounded searchers (`aesop` (Limperg & From, 2023), `rw_search`).

**Validation.** We evaluate RPE on 300 examples from the test set of xVerify (Chen et al., 2025). Detailed experiment setting and analysis are in Appendix F. RPE reaches 100% precision, 97.18% recall, and 0.9732 Cohen's kappa (Cohen, 1960). Rejected answers are mainly due to failure of xFinder and intolerance of floating-point errors. RPE ensures that the candidate $\hat{a}$ is not just mathematically equal, but semantically similar to the preferred solution $\bar{a}$.

## 4. Benchmarks

In this section, we present the construction of three benchmarks: **FormalMath500**, **MiniF2F-Solving**, and **PutnamBench-Solving**. Spanning difficulty levels from grade school mathematics to undergraduate competitions, these datasets serve as a testbed for the FPS and D-FPS frameworks. The current benchmarks focus on problems whose answers are unique or canonicalizable, which makes direct-answer comparison with RPE well defined.

### 4.1. Construction Pipeline

To bridge the gap between informal and formal reasoning, we construct data points as quadruples: (1) informal problem, (2) informal ground-truth, (3) formal problem $P = (V, \boxed{a}, \Phi, \Psi)$, and (4) formal ground-truth $\bar{a}$. This structure enables multi-faceted evaluation of informal reasoning, FTP, and our target tasks (FPS and D-FPS).

The benchmarks are constructed via a rigorous three-phase expert-in-the-loop pipeline:

**I. Problem Formalization.** The informal problem-answer pair is formalized into a Lean 4 statement $(\forall_{i=1}^{n}v_i, \bigwedge_{i=1}^{p}\phi_i \to \bigwedge_{i=1}^{q}\psi_i)$. Crucially, the answer is not hardcoded in the statement but left as a queriable free variable $\boxed{a}$.

**II. Logic Verification.** We employ off-the-shelf provers to detect low-quality formalizations via two checks. (1)

*Well-posedness.* We attempt to prove $\forall_{i=1}^n v_i, \bigwedge_{i=1}^p \phi_i \rightarrow$ `False`. If proven, the problem hypotheses are contradictory, indicating an invalid formalization. Human experts manually correct these formalizations; (2) *Soundness*. We attempt to prove that the formalized ground-truth satisfies the problem $(\forall_{i=1}^n v_i, \bigwedge_{i=1}^p \phi_i \rightarrow \bigwedge_{i=1}^q \psi_i)[\boxed{a} \mapsto \bar{a}]$. If not proven, the formalization is flagged as risky. 56.33% of FormalMath500, 54.67% of MiniF2F-Solving, and 1.23% of PutnamBench-Solving pass both checks.

**III. Manual Review.** Two experts review all formalizations to ensure semantic fidelity to the original informal problems. The revision rates are 46.77%, 22.93%, and 4.94% for FormalMath500, MiniF2F-Solving, and PutnamBench-Solving, respectively.

### 4.2. Benchmark Statistics

**FormalMath500** is a formalized subset of the prevalent MATH500 benchmark (Lightman et al., 2023). Problems in `Geometry` and `Counting & Probability` are excluded as their difficulty primarily stems from formalization rather than solving. The resulting benchmark includes 387 samples covering `Algebra` (123), `Intermediate Algebra` (92), `Number Theory` (62), `Prealgebra` (65), and `Precalculus` (45). The difficulty distribution is balanced across the 5 levels defined by (Hendrycks et al., 2021): Level 1 (38), Level 2 (67), Level 3 (87), Level 4 (96), and Level 5 (99).

**MiniF2F-Solving** is constructed by refactoring the MiniF2F benchmark (Zheng et al., 2022). While MiniF2F provides aligned informal and formal statements, many are originally problems ("Find $\boxed{a}$ s.t. $P(\boxed{a})$") cast as propositions ("Prove $P(\bar{a})$"). We reverse-engineer them by detaching the answer to restore the problem structure. Moreover, we address over-simplified formalizations, such as formalizing "Find the minimum of $f(x)$" (where the ground-truth is $c$) as merely "Prove $\forall x, f(x) \geq c$". In such cases, we restore the missing conditions to ensure semantic correctness. The refactored benchmark contains 375 samples, sourced from `AIME` (30), `MATH-Algebra` (140), `AMC` (82), `IMO` (3), and `MATH-Number Theory` (120).

**PutnamBench-Solving** is a refactored subset of PutnamBench (Tsoukalas et al., 2024), which consists of 644 propositions from undergraduate-level competitions. PutnamBench pioneered the use of `sorry` placeholders to factor out direct answers from propositions, facilitating our adaptation. After refactoring, the subset contains 324 samples. Topics include `Algebra` (138), `Analysis` (122), `Number Theory` (49), `Geometry` (28), `Linear Algebra` (25), `Combinatorics` (14), `Abstract Algebra` (9), `Probability` (8), and `Set Theory` (4).

## 5. Experiments

In this section, we evaluate existing methods for FPS and D-FPS on the three benchmarks. We present the experiment settings and analyze experimental results. For brevity, detailed hyperparameters and Lean 4 environment settings are deferred to Appendix I.1 and Appendix I.2, respectively.

### 5.1. Experiment Settings

**Evaluation Metrics.**

- **Solved%**: The success rate of finding a *correct* answer $\hat{a}$ such that $P(\hat{a})$ holds and $\hat{a} \approx_{\text{RPE}} \bar{a}$. This measures human-aligned problem solving using our proposed RPE.
- **Typecheck%**: The success rate of finding *any* answer $\hat{a}$ and proving $P(\hat{a})$. This measures pure formal soundness while disentangling alignment.
- **Proven%**: The success rate of proving the proposition $P(\bar{a})$ when the ground-truth answer $\bar{a}$ is explicitly provided (traditional FTP task).

**Evaluated Methods.**

- **FPS.** Given its similarity to FTP, we evaluate off-the-shelf FTP models in two paradigms: (1) ***Whole-Proof Generation*** (Xin et al., 2024b; Lin et al., 2025b; Wang et al., 2025; Ren et al., 2025; Lin et al., 2026; Dong & Ma, 2025), which outputs the full formal proof or solution directly; (2) ***Proof Search*** (Wu et al., 2025; Lin et al., 2025a), which navigates the proof state space with best-first search. We also evaluate DSP+ (Cao et al., 2026) and prompt generalist LLMs (DeepSeek-V3.2 (DeepSeek-AI et al., 2025b) and Claude Sonnet 4.5) as additional baselines.
- **D-FPS.** Due to its significant difference from FTP, we evaluate 10-shot prompting with generalist LLMs: (1) ***Direct***, directly generates forward solutions; and (2) ***Hybrid CoT***, interleaves informal reasoning with formal steps to leverage the model's chain-of-thought capabilities. Moreover, specialist LLM provers on D-FPS are evaluated in Appendix G.2, where they exhibit inferior performance and suffer from severe shortcut behavior.

### 5.2. Results and Discussions

Evaluation results of methods on FPS and D-FPS are summarized in Table 1, 2, respectively.

**Solving is harder than proving.** As shown in Table 1, across benchmarks and models, *Solved%* are consistently and significantly lower than *Proven%*. The best methods prove $80.19 \pm 0.15\%$ of FormalMath500, $86.22 \pm 0.15\%$ of MiniF2F-Solving, and $2.88 \pm 0.89\%$ of PutnamBench-Solving. However, the highest FPS solving rates are only $59.17 \pm 0.26\%$, $65.87 \pm 0.46\%$, and $1.03 \pm 0.18\%$.

*Table 1.* Evaluation results of methods on benchmarks formalized in FPS. *Solved* indicates both validity and alignment; *Typecheck* indicates only validity; *Proven* indicates proving the ground-truth answers' validity. Multi-run results are reported as mean±std.

| Benchmark | Method | Model | Proven% | Typecheck% | Solved% |
|---|---|---|---|---|---|
| Formal-Math500 | Proof Search | InternLM2.5-StepProver | 49.53±0.30% | 43.93±0.93% | 27.22±1.33% |
| | | LeanSTaR | 55.49±2.48% | 40.14±1.87% | 25.06±0.78% |
| | Whole-Proof Generation | DeepSeekProver-V1.5 | 47.89±0.54% | 9.13±0.39% | 8.01±0.26% |
| | | Goedel-Prover | 53.49±0.93% | 4.39±1.79% | 3.53±1.42% |
| | | Kimina-Prover | 60.38±0.98% | 31.52±0.45% | 30.32±0.15% |
| | | DeepSeek-Prover-V2 (non-CoT) | 68.91±0.65% | 0.43±0.39% | 0.43±0.39% |
| | | DeepSeek-Prover-V2 (CoT) | 72.18±1.04% | 34.80±0.65% | 33.68±1.04% |
| | | STP | 50.82±1.17% | 9.39±0.65% | 6.55±0.15% |
| | | Goedel-Prover-V2-8B | **80.19±0.15%** | **68.04±0.75%** | **59.17±0.26%** |
| | Prompt | DeepSeek-V3.2 | 39.79% | 18.35% | 14.99% |
| | | Sonnet-4.5 | 30.23% | 7.75% | 6.72% |
| | DSP+ | DeepSeek-V3.2+BFS-Prover-V1-7B | 20.93% | 30.49% | 28.42% |
| MiniF2F Solving | Proof Search | InternLM2.5-StepProver | 55.91±0.41% | 39.20±0.46% | 29.51±0.81% |
| | | LeanSTaR | 50.93±0.46% | 34.31±1.08% | 25.16±1.20% |
| | Whole-Proof Generation | DeepSeekProver-V1.5 | 59.11±1.01% | 1.24±0.81% | 0.80±0.46% |
| | | Goedel-Prover | 63.02±0.41% | 0.36±0.15% | 0.36±0.15% |
| | | Kimina-Prover | 72.80±0.27% | 32.00±0.46% | 31.91±0.62% |
| | | DeepSeek-Prover-V2 (non-CoT) | 75.56±0.62% | 0.09±0.15% | 0.09±0.15% |
| | | DeepSeek-Prover-V2 (CoT) | 78.40±0.71% | 22.67±1.87% | 22.40±1.62% |
| | | STP | 61.33±0.46% | 2.31±1.08% | 1.96±1.11% |
| | | Goedel-Prover-V2-8B | **86.22±0.15%** | **69.07±0.53%** | **65.87±0.46%** |
| | Prompt | DeepSeek-V3.2 | 42.40% | 17.87% | 16.27% |
| | | Sonnet-4.5 | 33.33% | 8.53% | 8.27% |
| | DSP+ | DeepSeek-V3.2+BFS-Prover-V1-7B | 23.47% | 23.20% | 21.60% |
| PutnamBench Solving | Proof Search | InternLM2.5-StepProver | 1.54±0.00% | 27.06±1.98% | 0.00±0.00% |
| | | LeanSTaR | 0.82±0.18% | 27.88±1.70% | 0.31±0.00% |
| | Whole-Proof Generation | DeepSeekProver-V1.5 | 0.93±0.00% | 7.51±0.64% | 0.10±0.18% |
| | | Goedel-Prover | 0.62±0.00% | 9.57±0.82% | 0.00±0.00% |
| | | Kimina-Prover | 1.54±0.00% | 12.14±0.47% | 0.93±0.00% |
| | | DeepSeek-Prover-V2 (non-CoT) | 1.95±0.47% | 4.84±0.36% | 0.00±0.00% |
| | | DeepSeek-Prover-V2 (CoT) | 2.67±0.18% | 12.96±0.00% | 0.62±0.00% |
| | | STP | 0.62±0.00% | 10.19±0.62% | 0.00±0.00% |
| | | Goedel-Prover-V2-8B | **2.88±0.89%** | 17.18±5.81% | **1.03±0.18%** |
| | Prompt | DeepSeek-V3.2 | 0.93% | 2.47% | 0.00% |
| | | Sonnet-4.5 | 0.62% | 0.31% | 0.00% |
| | DSP+ | DeepSeek-V3.2+BFS-Prover-V1-7B | 0.31% | 9.57% | 0.00% |

*Table 2.* Evaluation results of methods on benchmarks formalized in D-FPS. *Solved* indicates both validity and alignment; *Typecheck* indicates only validity.

| Benchmark | Method | Model | Typecheck% | Solved% |
|---|---|---|---|---|
| Formal-Math500 | Direct | DeepSeek-V3.2 | 30.49% | 29.46% |
| | Hybrid CoT | DeepSeek-V3.2 | **50.65%** | **48.58%** |
| | Direct | Sonnet-4.5 | 20.67% | 20.16% |
| | Hybrid CoT | Sonnet-4.5 | 24.03% | 23.26% |
| MiniF2F Solving | Direct | DeepSeek-V3.2 | 41.60% | 40.80% |
| | Hybrid CoT | DeepSeek-V3.2 | **58.93%** | **57.33%** |
| | Direct | Sonnet-4.5 | 28.53% | 28.00% |
| | Hybrid CoT | Sonnet-4.5 | 29.33% | 28.53% |
| PutnamBench Solving | Direct | DeepSeek-V3.2 | 0.93% | 0.00% |
| | Hybrid CoT | DeepSeek-V3.2 | **7.41%** | **1.54%** |
| | Direct | Sonnet-4.5 | 0.31% | 0.00% |
| | Hybrid CoT | Sonnet-4.5 | 0.00% | 0.00% |

**Off-the-shelf FTP models focus on validity at the expense of alignment.** Results of FPS reveal a systematic discrepancy between *Solved%* and *Typecheck%*. In FTP, the only goal is to construct any proof witness of a certain proposition. Once a term of the required type is constructed, the proof succeeds. In FPS, however, the answer term must also align with human preferences, e.g., simplest form, direct numerical value. That is to say, tautologically answering "find all $x$ s.t. $x^2 - 1 = 0$" with $\{x \mid x^2 - 1 = 0\}$ is not allowed. Manual inspection shows that $98.7\%$ of typechecked-but-unsolved cases are true negatives; among them, $10.3\%$ are unsimplified terms and $89.7\%$ are tautological answers. The requirement of a human-aligned answer constitutes an "alignment tax" when moving from proving to solving.

**D-FPS improves alignment but suffers from under-training.** Table 2 shows that *Typecheck%* is generally close to *Solved%* for generalist LLMs under D-FPS, which further demonstrates D-FPS's effectiveness in alleviating "cheating". Hybrid CoT improves over Direct prompting for DeepSeek-V3.2, reaching $48.58\%$, $57.33\%$, and $1.54\%$ solved rates on the three benchmarks. Moreover, preference model experiments in Appendix G.1 show that formal solutions in D-FPS are more aligned with human preference than FPS (avg. $> 0.75$ across two preference models). However, current prompting-based D-FPS methods struggle to solve difficult problems. This is expected because existing pre-training and SFT data of formal verification focuses on FTP, and the paradigm shift from FTP to D-FPS is significantly more severe than to FPS.

**Proof failure hinders simple problems, while reasoning flaws dominate hard ones.** We provide a detailed analysis of the D-FPS error modes in Table 3. For easier benchmarks, "Solution" errors (Lean-check failures in the generated formal solution) account for about 70% of failures, indicating that models often identify the correct answer but lack the formal rigor to prove it. In contrast, for the undergraduate-level PutnamBench-Solving, "Answer" errors (wrong answer) and "Submission" errors (irrelevant outputs) prevail, reflecting fundamental difficulties in solving complex prob-

*Table 3.* Error distribution of D-FPS experiments. **Length** means incomplete responses truncated by token limits; **Rejection** means API refusal; **Format** means unparsable responses; **Submission** means the submitted answer is irrelevant to the queriable; **Answer** means the informal answer is wrong; **Solution** means the generated formal solution fails the Lean 4 check.

| Benchmark | Method | Length | Rejection | Format | Submission | Answer | Solution |
|---|---|---|---|---|---|---|---|
| **Formal-Math500** | Direct | 0.21% | 0.00% | 0.22% | 1.07% | 26.37% | 72.14% |
| | Hybrid CoT | 3.31% | 0.02% | 0.60% | 1.82% | 26.22% | 68.03% |
| **MiniF2F Solving** | Direct | 0.32% | 0.00% | 0.28% | 0.85% | 20.26% | 78.28% |
| | Hybrid CoT | 4.57% | 0.00% | 0.64% | 1.79% | 21.77% | 71.24% |
| **PutnamBench Solving** | Direct | 0.50% | 0.00% | 1.60% | 8.47% | 64.70% | 24.73% |
| | Hybrid CoT | 5.71% | 0.00% | 0.95% | 22.76% | 54.91% | 15.67% |

lems. Hybrid CoT shows a slight increase in length-related errors due to verbose reasoning but generally shares this distribution with Direct.

## 6. Conclusion

This work extends the frontier of formal reasoning from *verification* (proving known propositions) to *problem-solving* (discovering unknown answers). We establish **FPS** (*Formal Problem-Solving*) to unify answer discovery and proof construction, and propose **D-FPS** (*Deductive FPS*) to structure solving into forward derivation and backward verification. To facilitate this paradigm shift, we contribute three high-quality benchmarks spanning difficulty levels from grade school to undergraduate competitions: **FormalMath500**, **MiniF2F-Solving**, and **PutnamBench-Solving**. Moreover, we introduce **RPE** (*Restricted Propositional Equivalence*), a more expressive and faithful semantic evaluation metric. Extensive experiments across specialist FTP methods and prompting generalist LLMs reveal that solving is significantly more challenging than proving, confirming the necessity of dedicated solving benchmarks. They also expose an "alignment tax" when transitioning from FTP's type-oriented proof construction to FPS's human-aligned answer construction.

Discussions of limitations and future work are deferred to Appendix J.

## Impact Statement

Our research focuses on performing and evaluating process-verified problem-solving. While our methods offer potential benefits for the trustworthiness and verifiability of AI reasoning, we acknowledge the importance of considering the ethical implications of deploying such models. These include ensuring the responsible use of LLMs, mitigating biases in model outputs, and addressing accessibility concerns. We commit to making our code available for transparency and encourage the community to use our findings responsibly, considering the societal impacts of deploying LLMs.

The three benchmarks are annotated by the authors based on existing datasets:

- **FormalMath500.** The informal problems and answers are sampled from the MATH (Hendrycks et al., 2021) dataset, which adopts the permissive MIT License;

- **MiniF2F-Solving.** The original formal propositions are from the MiniF2F (Zheng et al., 2022) benchmark, whose Lean subset is released under the permissive Apache 2.0 License;

- **PutnamBench-Solving.** The original formal propositions are from the PutnamBench (Tsoukalas et al., 2024) benchmark, whose Lean subset is released under the permissive Apache 2.0 License;

We believe our use of the aforementioned datasets aligns with their intended purpose. We fully respect and acknowledge their authors' valuable work and outstanding contributions to the community. The released artifacts include the original copyright notices, license statements, disclaimers, and notices of any modifications made.

The proposed benchmarks focus on formal mathematical problem-solving and do not contain sensitive or personal information.

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

# A. Background

## A.1. Formal Theorem Proving

Based on the Curry-Howard isomorphism (Howard et al., 1980) (a.k.a. propositions-as-types correspondence), interactive proof assistants such as Lean (Moura & Ullrich, 2021) and Coq (Barras et al., 1997) can verify proofs of mathematical theorems and assertions about complex systems by type checking the given terms.

To construct such a "proof term", one can directly construct a term of the corresponding proposition's type. This direct proof is called a "term-style" proof (Avigad et al., 2024). Another incremental approach is a "tactic-style" proof (Avigad et al., 2024) (a.k.a. proof script), a series of tactic applications that continually reduce goals into subgoals until all goals are resolved.

In tactic mode, proof assistants maintain a *proof state*, a set of proof goals. Each *goal* $\Gamma \vdash U$ contains a *local context* $\Gamma$, which is a telescope (ordered list) of declarations, and a *conclusion* $U$, which is a type to construct. Each *declaration* might be a *local assumption* $(x : T)$ (variable declaration or hypothesis), or a *local definition* $(x := t : T)$. A *tactic* is a partial function that manipulates the current proof state by closing a goal $\Gamma \vdash \phi$ and creating a finite set of subgoals $\{\Gamma'_i \vdash U'_i\}_{i=1}^n$ ($n$ can be 0). Tactic applications are provability-reflecting: if all subgoals $\{\Gamma'_i \vdash U'_i\}_{i=1}^n$ are solved, then the original goal $\Gamma \vdash \phi$ is solved (Aniva et al., 2024).

$$\frac{\begin{array}{c} \Gamma'_1 \vdash U'_1 \\ \Gamma'_2 \vdash U'_2 \\ \cdots\cdots \\ \Gamma'_n \vdash U'_n \end{array}}{\Gamma \vdash C} \ \text{PR}$$

Some "safe[4] tactics" are further provability-preserving, i.e. if the original goal $\Gamma \vdash \phi$ is provable, then all subgoals $\{\Gamma'_i \vdash U'_i\}_{i=1}^n$ are also provable.

$$\frac{\Gamma \vdash C}{\begin{array}{c} \Gamma'_1 \vdash U'_1 \\ \Gamma'_2 \vdash U'_2 \\ \cdots\cdots \\ \Gamma'_n \vdash U'_n \end{array}} \ \text{PP}$$

When sufficient information is unavailable for tactic applications, metavariables are introduced to represent "holes" in expressions. Unresolved proof goals are similarly represented by metavariables internally (Community, 2025). A *metavariable* is a typed placeholder that represents the same expression in all occurrences. It carries a local context (same as that of goals) and a target type (corresponding to the conclusions of goals). A metavariable must be assigned an expression of its target type, using only free variables from its local context and information from the global context.

## A.2. Equality

In Lean 4, there are three levels of equivalence between terms (Buzzard, 2024):

1. **Syntactic equality** is the strongest kind of equivalence. Two terms are *syntactically equal* if they have the same syntactic structures. For example, `x␣+␣0` is syntactically equal to `x+␣␣0` because redundant whitespaces[5] are syntactically neglected in this case, while `x  +  0` and `x` are not syntactically equal since "`+  0`" introduces a syntactic difference.

2. **Definitional equality** is a relatively weaker kind of equivalence. Two terms are *definitionally equal* if they are convertible under a series of conversion rules, including $\alpha$-*conversion*, $\eta$-*expansion*, *proof irrelevance*, $\beta$-*reduction*, etc. (Bailey et al., 2024). Although weaker than syntactic equality, definitional equality remains too strong, even for determining equivalence between propositions (Liu et al., 2025). For instance, $2 + 1$ is not definitionally equal to $1 + 2$, where 1 and 2 are real numbers.

---

[4] Named by (Limperg & From, 2023).
[5] Here "␣" is used to emphasize whitespaces.

3. **Propositional equality** is the weakest kind of equivalence. Two terms $t$ and $t'$ are *propositionally equal* if the proposition $t = t'$ is provable, i.e., a proof term of type $t = t'$ can be constructed. However, this form of equality is too weak when determining equivalence between an answer term and the ground truth. For example, in the find-all problem "Find all $x$ of type $T_x$ such that $P\ x$," with the ground truth $\bar{S}$ of type `Set` $T_x$, a direct adaptation of the problem $\{x : T_x \mid P\ x\}$ is propositionally equal to $\bar{S}$.

## B. More Related Works

**Philosophical Discussions.** Since the proposal of erotetic logic (Collingwood, 2015), also known as the logic of questions and answers (Belnap et al., 1976), a significant body of discussions has emerged concerning questions, answers, and their relations from the perspectives of philosophy, linguistics, and logic. Within their consensus, questions are categorized into *elementary questions* (including *whether-questions* and *which-questions*), *why-questions*, and *embedded questions* (a.k.a. indirect questions).

The semantics of elementary questions can be broadly summarized into four theories (Cross & Roelofsen, 2024). Among these, Hamblin Semantics, Partition Semantics, and Inquisitive Semantics are proposition-set theories, as they conceptualize the meaning of questions as sets of propositions. In Hamblin Semantics (Hamblin, 1973; Karttunen, 1977; Belnap et al., 1976), a question is defined as a function that maps a possible world into a set of propositions, each corresponding to a possible answer, where a possible world refers to a complete and consistent way how things could have been (Menzel, 2024). (Nickles, 1981) further defines a problem as the constraints on the solution and the requirement that a solution exists. However, Hamblin Semantics lacks a clear definition for "what a possible answer should be." To solve this limitation, Partition Semantics (Groenendijk & Stokhof, 1982; Aloni et al., 2007) restricts the set of propositions mapped by a question to be mutually exclusive and exhaustive over the entire logical space. The resulting definition of possible answers is a "true exhaustive answer." However, in some cases, the "true exhaustive answer" is challenging to specify. Another attempt to clarify the definition of possible answers is Inquisitive Semantics (Groenendijk, 2007; Ciardelli et al., 2013; 2017), which extends classical logic by incorporating questions and interrogatives, treating both questions and propositions as fundamental concepts. This line of theories defines the meaning of questions as downward-closed proposition sets.

Our work differs by formalizing both whether- and which-questions and their solving processes from the perspective of formal verification. This enables seamless integration with existing theorem proving environments.

**Evaluation of Answer Correctness.** Existing methods are all designed for informal problem-solving. They evaluate by string matching (Cobbe et al., 2021; Hendrycks et al., 2021; Li et al., 2024a), symbolic equivalence (Lewkowycz et al., 2022; Lightman et al., 2023; Glazer et al., 2024) via SymPy (Meurer et al., 2017), or domain-specific LLMs (Chen et al., 2025). However, these methods may lack numerical robustness (e.g., false positive between $0.999999997$ and $1$), fall short on complex answers (e.g., a false negative between $\{x \in \mathbb{R} \mid 0 \le x \le 1\}$ and $[0, 1]$), or rely on LLMs. In contrast, we propose a symbolic approach based on formal verification, providing expressive, interpretable, human-aligned, and lightweight correctness checking.

Recent work on statement autoformalization (Liu et al., 2025) introduces a neuro-symbolic equivalence relation between formal statements by checking bidirectional implication. RPE targets a different object: it evaluates propositional equality between answer objects in FPS/D-FPS. It also uses a fixed symbolic tactic set, reducing LLM overhead and making the evaluation deterministic.

## C. Exemplar Reasoning Patterns

Many reasoning patterns in problem-solving can be unified under this formulation, including:

- Deriving a new condition by deductive reasoning on conditions $\Phi_i$. This step adds one new condition $\phi'_i$ to $\Phi_i$ if $\bigwedge_{j=1}^{p_{i,j}} \phi_{i,j} \to \phi'_i$;

- Deducing necessary conditions of the answer by partial deductive reasoning (Xie et al., 2024b) on the conditions $\Phi$ and conclusions $\Psi$; This step adds one new condition $\bigwedge_{j=1}^{q_{i,j}} \psi_{i,j} \to \phi'_i$ to $\Phi_i$ if $(\bigwedge_{j=1}^{p_{i,j}} \phi_{i,j} \wedge \bigwedge_{j=1}^{q_{i,j}} \psi_{i,j}) \to \phi'_i$;

- Drawing sufficient conditions of conclusions $\Psi$ by backward reasoning. This step replaces one conclusion $\psi_{i,j}$ to $\psi'_{i,j}$ if $\psi'_{i,j} \to \psi_{i,j}$;

- Case-by-case discussion. This step replaces one goal with multiple goals and adds one concrete condition to each goal;

- Extracting a new hole from one $\exists$-quantified conclusion. This step replaces the quantified variable with a newly introduced hole;

- Filling the hole $a$ with a term and proposing possibly other holes. This step replaces every occurrence of the hole into a term and removes the hole.

## D. Exemplar Formalizations of Problems

### D.1. Exemplar Problem in Different Types

1. *Yes-no question*: Does there exist a positive real number $\alpha$ such that $[\alpha^n] - n$ is even for all integers $n > 0$?
   The formalization is straightforward:

   $V = \emptyset, \Phi = \emptyset, \hat{a} := \text{True}$, and
   $$\Psi = \{\boxed{a} \in \{\text{True}, \text{False}\},$$
   $$(\boxed{a} \leftrightarrow \exists \alpha, \forall n \in \mathbb{Z}, n > 0 \rightarrow ([\alpha^n] - n)$$
   $$:= 0 \bmod 2)\}$$

   The code implementation of the FPS framework is:

   ```
   problem YesNo
   find (a : Prop) s.t.
   : a ↔ ∃ α : ℝ, ∀ n : ℕ, n > 0 → (α^n - n) % 2 = 0 := solve
   ```

   The code implementation of the D-FPS framework is:

   ```
   problem YesNo_D
   find_all (a : Prop) iff
   : a ↔ ∃ α : ℝ, ∀ n : ℕ, n > 0 → (α^n - n) % 2 = 0 := solve
   ```

2. *Equation solving*: Solve the equation $x^2 - 1 = 0$.
   The formalization depends on the concrete meaning of the problem: find one answer or find all possible answers. The find-one problem can be formulated as

   $$V = \emptyset, \Phi = \emptyset, \hat{a} := -1 \vee \hat{a} := 1$$
   $$\Psi = \{\boxed{a} \in \mathbb{R}, \boxed{a}^2 - 1 = 0\}$$

   The code implementation of the FPS framework is:

   ```
   problem Eq_One
   find (a : ℝ) s.t.
   : a^2 - 1 = 0 := solve
   ```

   Notably, the D-FPS framework is unsuitable for the find-one problem so we omit the code implementation of the D-FPS framework here.

   The find-all problem can be

   $$V = \{x\}, \Phi = \{x \in \mathbb{R}\}, \hat{a} := \{-1, 1\}$$
   $$\Psi = \{\boxed{a} \in 2^{\mathbb{R}}, x \in \boxed{a} \leftrightarrow x^2 - 1 = 0\}$$

   The code implementation of the FPS framework is:

   ```
   problem Eq_All
   (x : ℝ)
   find (a : Set ℝ) s.t.
   : x ∈ a ↔ x^2 - 1 = 0 := solve
   ```

   The code implementation of the D-FPS framework is:

   ```
   problem Eq_All_D
   find_all (x : ℝ) iff
   : x^2 - 1 = 0 := solve
   ```

3. *Simplification*: Simplify $\sqrt{28x} \cdot \sqrt{15x} \cdot \sqrt{21x}$.
   This problem is somewhat ill-defined since the metric of "simplify" is not clear[6]. But it can still be formulated as follows:

   $$V = \{x\}, \Phi = \{x \in \mathbb{R}^+\}, \hat{a} := 42x\sqrt{5x}$$
   $$\Psi = \{\boxed{a} \in \mathbb{R}, \boxed{a} = \sqrt{28x} \cdot \sqrt{15x} \cdot \sqrt{21x}\}$$

   The code implementation of the FPS framework is:

```
problem Simp
(x : ℝ)
find (a : ℝ) s.t.
(h_1 : x > 0)
: a = sqrt(28 * x) * sqrt(15 * x) * sqrt(21 * x) := solve
```

   The code implementation of the D-FPS framework is:

```
problem Simp_D
(x : ℝ)
find_all (a : ℝ) iff
(h_1 : x > 0)
: a = sqrt(28 * x) * sqrt(15 * x) * sqrt(21 * x) := solve
```

4. *Counter-example construction*: Find a Fermat number $F_n = 2^{2^n} + 1$ which is not prime.
   One possible answer is 5. ($F_5 = 4,294,967,297 = 641 \times 6,700,417$)
   Let Prime be a predicate that a given natural number is a prime number, the formalization is

   $$V = \emptyset, \Phi = \emptyset, \hat{a} := 5 \wedge \ldots$$
   $$\Psi = \{\boxed{a} \in \mathbb{N}, \neg\text{Prime}(2^{2^{\boxed{a}}} + 1)\}$$

   The code implementation of the FPS framework is:

```
problem Counter
find (a : ℕ) s.t.
: ¬ Nat.Prime (2^(2^a) + 1) := solve
```

   Notably, the D-FPS framework is unsuitable for the find-one problem so we omit the code implementation of the D-FPS framework here.

5. *Physics Modeling*: A spherical droplet falls through the stationary mist, absorbing all encountered water molecules. Assuming the droplet remains spherical, mist density is uniform, air viscosity is negligible, and gravitational acceleration $g$ is constant. After sufficient time, the droplet's acceleration approaches a steady value. Determine this value.
   The answer is $\frac{1}{7}g$. Interestingly, it does not depend on variables such as the mist density or the density of water.
   We denote the mass of the droplet by $m$, the velocity by $v$, the radius by $R$, the density of water by $\rho$, and the mass density of the mist by $k$. Then $V = \{g, \rho, k, t, m, v, R\}$ The physical process can be modeled as

   $$\Phi = \begin{cases} g \in \mathbb{R}^+, \rho \in \mathbb{R}^+, k \in \mathbb{R}^+, m \in \mathbb{R}^{*\mathbb{R}}, \\ v \in \mathbb{R}^{*\mathbb{R}}, R \in \mathbb{R}^{*\mathbb{R}}, m(-\infty) = 0, \\ v(-\infty) = 0, mg = m\frac{\mathrm{d}v}{\mathrm{d}t} + v\frac{\mathrm{d}m}{\mathrm{d}t}, \\ m = \frac{4}{3}\pi R^3 \rho, \frac{\mathrm{d}m}{\mathrm{d}t} = k\pi R^2 v \end{cases}$$

   And $\Psi = \{\boxed{a} \in \mathbb{R}^*, \boxed{a} = (\frac{\mathrm{d}v}{\mathrm{d}t})(\infty)\}, \hat{a} = \frac{1}{7}g$

   The code implementation of the FPS framework is:

---

[6]RPE in Sec. 3.4 can provide a human-aligned evaluation for direct answers.

```
problem Physics
(g : ℝ) (ρ : ℝ) (k : ℝ) (t : ℝ) (m : ℝ → ℝ) (v : ℝ → ℝ) (R : ℝ → ℝ)
find (a : ℝ) s.t.
(h_1 : g > 0) (h_2 : ρ > 0) (h_3 : k > 0)
(h_4 : Tendsto (m t) atBot (𝒩 0))
(h_5 : Tendsto (v t) atBot (𝒩 0))
(h_6 : (m t) * g = (m t) * (deriv v t) + (v t) * (deriv m t))
(h_7 : m t = 4/3 * Real.pi * (R t)^3 * (ρ t))
(h_8 : deriv m t = k * Real.pi * (R t)^2 * (v t))
: Tendsto (v t) atTop (𝒩 a) := solve
```

The code implementation of the D-FPS framework is:

```
problem Physics_D
(g : ℝ) (ρ : ℝ) (k : ℝ) (t : ℝ) (m : ℝ → ℝ) (v : ℝ → ℝ) (R : ℝ → ℝ)
find_all (a : ℝ) iff
(h_1 : g > 0) (h_2 : ρ > 0) (h_3 : k > 0)
(h_4 : Tendsto (m t) atBot (𝒩 0))
(h_5 : Tendsto (v t) atBot (𝒩 0))
(h_6 : (m t) * g = (m t) * (deriv v t) + (v t) * (deriv m t))
(h_7 : m t = 4/3 * Real.pi * (R t)^3 * (ρ t))
(h_8 : deriv m t = k * Real.pi * (R t)^2 * (v t))
: Tendsto (v t) atTop (𝒩 a) := solve
```

## D.2. Exemplar Problem in the Benchmarks

### FormalMath500

1. How many integers are in the solution set of $|x - 2| \leq 5.6$?

   The problem can be formulated as follows:

   $$V = \{S\}, \Phi = \{S \in 2^{\mathbb{Z}}, S = \{x : \mathbb{Z} \mid |x - 2| \leq 28/5\}\}, \hat{a} := 11$$
   $$\Psi = \{\boxed{a} \in \mathbb{N}, |S| = \boxed{a}\}$$

   The code implementation of the FPS framework is:

```
problem Q111
find (answer : ℕ) s.t.
  (S : Set ℤ)
  (hS : S = {x : ℤ | abs (x - 2) ≤  (28 : ℚ) / (5 : ℚ) })
: (S.encard = answer)
:= solve
```

   The code implementation of the D-FPS framework is:

```
problem Q111
find_all (answer : ℕ) iff
  (S : Set ℤ)
  (hS : S = {x : ℤ | abs (x - 2) ≤  (28 : ℚ) / (5 : ℚ) })
: (S.encard = answer)
:= solve
```

2. The proper divisors of 12 are 1, 2, 3, 4, and 6. A proper divisor of an integer $N$ is a positive divisor of $N$ that is less than $N$. What is the sum of the proper divisors of the sum of the proper divisors of 284?

   The problem can be formulated as follows:

   $$V = \emptyset, \Phi = \emptyset, \hat{a} := 220$$
   $$\Psi = \{\boxed{a} \in \mathbb{N}, \sum_{i|284, i<284} i = \boxed{a}\}$$

   The code implementation of the FPS framework is:

```
problem Q234
find (answer : ℕ) s.t.
   (sum : ℕ)
   (h_sum : sum = Σ x in Nat.properDivisors 284, x)
: (answer = Σ d in Nat.properDivisors sum, d)
:= solve
```

The code implementation of the D-FPS framework is:

```
problem Q234
find_all (answer : ℕ) iff
   (sum : ℕ)
   (h_sum : sum = Σ x in Nat.properDivisors 284, x)
: (answer = Σ d in Nat.properDivisors sum, d)
:= solve
```

## MiniF2F-Solving

1. Define a function on the positive integers recursively by $f(1) = 2$, $f(n) = f(n-1) + 1$ if $n$ is even, and $f(n) = f(n-2) + 2$ if $n$ is odd and greater than 1. What is $f(2017)$?

   The problem can be formulated as follows:

   $$V = \{f\}, \Phi = \{f \in \mathbb{R}^{\mathbb{N}}, f(1) = 2, (\forall\, n \in \mathbb{N}, (1 < n \land \text{Even } n) \to f(n) = f(n-1) + 1),$$
   $$(\forall\, n \in \mathbb{N}, (1 < n \land \text{Odd } n) \to f(n) = f(n-2) + 2)\}, \hat{a} := 2018,$$
   $$\Psi = \{\boxed{a} \in \mathbb{R}, \boxed{a} = f(2017)\}$$

   The code implementation of the FPS framework is:

```
problem Q68
find (answer : ℝ) s.t.
   (f : ℕ → ℝ)
   (h₀ : f 1 = 2)
   (h₁ : ∀ n, 1 < n ∧ Even n → f n = f (n - 1) + 1)
   (h₂ : ∀ n, 1 < n ∧ Odd n → f n = f (n - 2) + 2)
: (answer = f 2017)
:= solve
```

   The code implementation of the D-FPS framework is:

```
problem Q68
find_all (answer : ℝ) iff
   (f : ℕ → ℝ)
   (h₀ : f 1 = 2)
   (h₁ : ∀ n, 1 < n ∧ Even n → f n = f (n - 1) + 1)
   (h₂ : ∀ n, 1 < n ∧ Odd n → f n = f (n - 2) + 2)
: (answer = f 2017)
:= solve
```

2. Find the units digit of $16^{17} \times 17^{18} \times 18^{19}$.

   The problem can be formulated as follows:

   $$V = \emptyset, \Phi = \emptyset, \hat{a} := 8,$$
   $$\Psi = \{\boxed{a} \in \mathbb{N}, \boxed{a} = (16^{17} \times 17^{18} \times 18^{19})\%10\}$$

   The code implementation of the FPS framework is:

```
problem Q308
find (answer : ℕ) s.t.
: (answer = (16^17 * 17^18 * 18^19) % 10)
:= solve
```

The code implementation of the D-FPS framework is:

```
problem Q308
find_all (answer : ℕ) iff
: (answer = (16^17 * 17^18 * 18^19) % 10)
:= solve
```

## PutnamBench-Solving

1. Evaluate $\int_0^1 \frac{\ln(x+1)}{x^2+1}\,dx$.

   The problem can be formulated as follows:

   $$V = \emptyset, \Phi = \emptyset, \hat{a} := \pi \log 2/8,$$

   $$\Psi = \{\boxed{a} \in \mathbb{R}, \int_0^1 \frac{\ln(x+1)}{x^2+1}\,dx = \boxed{a}\}$$

   The code implementation of the FPS framework is:

   ```
   problem Q198
   find (answer : ℝ) s.t.
   : (∫ x in (0:ℝ)..1, (Real.log (x+1))/(x^2 + 1) = answer)
   := solve
   ```

   The code implementation of the D-FPS framework is:

   ```
   problem Q198
   find_all (answer : ℝ) iff
   : (∫ x in (0:ℝ)..1, (Real.log (x+1))/(x^2 + 1) = answer)
   := solve
   ```

2. Find

   $$\sum_{i=1}^{\infty}\sum_{j=1}^{\infty} \frac{1}{i^2 j + 2ij + ij^2}.$$

   The problem can be formulated as follows:

   $$V = \emptyset, \Phi = \emptyset, \hat{a} := 7/4,$$

   $$\Psi = \{\boxed{a} \in \mathbb{Q}, \sum_{i=1}^{\infty}\sum_{j=1}^{\infty} \frac{1}{i^2 j + 2ij + ij^2} = \boxed{a}\}$$

   The code implementation of the FPS framework is:

   ```
   problem Q53
   find (answer : ℚ) s.t.
   : ((Σ' i : ℕ+, Σ' j : ℕ+, (1 : ℚ) / (i ^ 2 * j + 2 * i * j + i * j ^ 2)) = answer)
   := solve
   ```

   The code implementation of the D-FPS framework is:

   ```
   problem Q53
   find_all (answer : ℚ) iff
   : ((Σ' i : ℕ+, Σ' j : ℕ+, (1 : ℚ) / (i ^ 2 * j + 2 * i * j + i * j ^ 2)) = answer)
   := solve
   ```

# E. Proofs of Properties

## E.1. Soundness of FPS

**Theorem.** *FPS is **sound**: for any problem $P$ and direct answer $\hat{a}$ produced by FPS, $P(\hat{a})$ holds.*

*Proof.* The Lean 4 proof state of FPS initializes as:

```
case Proof
(v₁ : T₁)...(vₙ : Tₙ)
(h₁ : φ₁)···(hₚ : φₚ)
⊢ (ψ₁ ∧ ··· ∧ ψ_q)[ a ↦ ?Answer]
case Answer
(v₁ : T₁)...(vₙ : Tₙ)
⊢ T_a
```

Therefore, upon finishing the proof, we can extract from the Lean 4 kernel:

- A term $\hat{a} : T_a$ filling the metavariable `?Answer` from the initial goal `case Answer`

- A proof term $h$ from the initial goal `case Proof`.

Since `?Answer` is filled by $\hat{a}$, $h$ is actually a proof of $(\forall_{i=1}^{n}(v_i : T_i), \forall_{i=1}^{p}(h_i : \phi_i), (\wedge_{i=1}^{q}\psi_i)[\boxed{a} \mapsto \hat{a}]$, i.e. $P(\hat{a})$.

Therefore, $P(\hat{a})$ holds by the proof term $h$. $\qquad\square$

## E.2. Expressiveness of D-FPS for Find-All Problems

**Theorem.** *Regarding find-all problems, the expressiveness of D-FPS is at least as strong as that of FPS.*

*Proof.* We construct an injection from an arbitrary find-all FPS problem to D-FPS while preserving semantics.

Suppose the FPS problem consists of $(V, (\boxed{a} : T_a), \Phi, \Psi)$ and the ground-truth answer is $\bar{a}$. We have

$$\forall_{i=1}^{n}(v_i : T_i), \forall_{i=1}^{p}(h_i : \phi_i), (\wedge_{i=1}^{q}\psi_i)[\boxed{a} \mapsto \bar{a}]$$

Therefore, the following assertion holds.

$$\forall_{i=1}^{n}(v_i : T_i), \forall_{i=1}^{p}(h_i : \phi_i), \forall(\boxed{a} : T_a), (\boxed{a} = \bar{a}) \rightarrow (\wedge_{i=1}^{q}\psi_i) \qquad (3)$$

Since it is a find-all problem, $\bar{a}$ is the only answer (for find-unique-one problems) or the complete collection of all valid answers (for multiple-answer problems). The following assertion holds.

$$\forall_{i=1}^{n}(v_i : T_i), \forall_{i=1}^{p}(h_i : \phi_i), \forall(\boxed{a} : T_a), (\wedge_{i=1}^{q}\psi_i) \rightarrow (\boxed{a} = \bar{a}) \qquad (4)$$

Therefore, composing Proposition 3 and Proposition 4, the following proposition holds:

$$\forall_{i=1}^{n}(v_i : T_i), \forall_{i=1}^{p}(h_i : \phi_i), \forall(\boxed{a} : T_a), (\wedge_{i=1}^{q}\psi_i) \leftrightarrow (\boxed{a} = \bar{a})$$

i.e.

$$\forall_{i=1}^{n}(v_i : T_i), \forall(\boxed{a} : T_a), \forall_{i=1}^{p}(h_i : \phi_i), ((\wedge_{i=1}^{q}\psi_i) \leftrightarrow \boxed{A})[\boxed{A} \mapsto a = \bar{a}]$$

which corresponds to the D-FPS problem $(V \cup \{(\boxed{a} : T_a)\}, (\boxed{A} : \texttt{Prop}), \Phi, \{\bigwedge_{i=1}^{q}\psi_i \leftrightarrow \boxed{A}\})$ with ground-truth answer $\bar{A} = (\boxed{a} = \bar{a})$. $\qquad\square$

Specifically, a multiple-answer problem with ground-truth answer $\bar{a}$ formulated in FPS as $(V, (\boxed{a} : \texttt{Set } T_x), \Phi, \{\boxed{a} = \{x : T_x | \bigwedge_{i=1}^{q}\psi_i\}\})$ or $(V \cup \{(x : T_x)\}, (\boxed{a} : \texttt{Set } T_x), \Phi, \{\bigwedge_{i=1}^{q}\psi_i \leftrightarrow x \in \boxed{a}\})$ can be mapped into D-FPS as $(V \cup \{(x : T_x)\}, (\boxed{A} : \texttt{Prop}), \Phi, \{\bigwedge_{i=1}^{q}\psi_i \leftrightarrow \boxed{A}\})$ with ground-truth answer $\bar{A} := x \in \bar{a}$ (neither $\boxed{a}$ nor $x$ occurs free in $\Phi$).

### E.3. Completeness of D-FPS for Find-All Problems

**Theorem.** *D-FPS is **complete**: for any find-all problem with ground-truth $\bar{A}$, for any direct answer $\hat{A}$ resulted from D-FPS, the following proposition holds:*

$$\forall_{i=1}^{n}(v_i : T_i), \forall_{i=1}^{p}(h_i : \phi_i), \bar{A} \rightarrow \hat{A}$$

*Proof.* Suppose the D-FPS problem consists of $(V, (\boxed{A} : \texttt{Prop}), \Phi, \{\psi \leftrightarrow \boxed{A}\})$ and the Lean 4 proof state is initialized as:

```
case Forward
(v₁ : T₁)...(vₙ : Tₙ)
(h₁ : φ₁)···(hₚ : φₚ)
(h_c : ψ)
⊢? A
case Backward
(v₁ : T₁)...(vₙ : Tₙ)
(h₁ : φ₁)···(hₚ : φₚ)
(h_a :  ? A )
⊢ ψ
case Answer
(v₁ : T₁)...(vₙ : Tₙ)
⊢ Prop
```

Therefore, upon finishing forward-solving, we can extract from the Lean 4 kernel:

- A term $\hat{A} : \texttt{Prop}$ filling the metavariable `?Answer` from the initial goal `case Answer`

- A proof term $h^{\rightarrow}$ from the initial goal `case Forward`.

Since `?Answer` is filled by $\hat{A}$, $h^{\rightarrow}$ is actually a proof of

$$\forall_{i=1}^{n}(v_i : T_i), \forall_{i=1}^{p}(h_i : \phi_i), \forall(h_c : \psi), \hat{A}$$

The ground-truth $\bar{A}$ satisfies

$$\forall_{i=1}^{n}(v_i : T_i), \forall_{i=1}^{p}(h_i : \phi_i), (\psi \leftrightarrow \bar{A})$$

Therefore, we have

$$\forall_{i=1}^{n}(v_i : T_i), \forall_{i=1}^{p}(h_i : \phi_i), \forall(h' : \bar{A}), \hat{A}$$

i.e.,

$$\forall_{i=1}^{n}(v_i : T_i), \forall_{i=1}^{p}(h_i : \phi_i), \bar{A} \rightarrow \hat{A}$$

$\square$

### E.4. Soundness of D-FPS for Find-All Problems

**Theorem.** *D-FPS is **sound**: for any find-all problem with ground-truth $\bar{A}$, for any direct answer $\hat{A}$ resulted from D-FPS, if the backward-proving is finished, the following proposition holds:*

$$\forall_{i=1}^{n}(v_i : T_i), \forall_{i=1}^{p}(h_i : \phi_i), \hat{A} \rightarrow \bar{A}$$

*Proof.* Upon finishing backward-proving, apart from $\hat{A}$, we can extract a proof term $h^{\leftarrow}$ from the initial goal `case Backward`.

Since `?Answer` is filled by $\hat{A}$, $h^{\leftarrow}$ is actually a proof of

$$\forall_{i=1}^{n}(v_i : T_i), \forall_{i=1}^{p}(h_i : \phi_i), (\forall(h_a :? \boxed{A}), \psi)[? \boxed{A} \mapsto \hat{A}]$$

*Table 4.* Ablation of the restricted tactic set in RPE.

| Tactic Set | Precision | Recall | Cohen's Kappa |
|---|---|---|---|
| Only `rw_search` | 100.00% | 78.17% | 0.7904 |
| +`rfl` | 100.00% | 81.69% | 0.8245 |
| +`norm_num` | 100.00% | 95.77% | 0.9598 |
| +`aesop` | 100.00% | 96.48% | 0.9665 |
| +`ring_nf` (Current) | 100.00% | 97.18% | 0.9732 |

i.e.,

$$\forall_{i=1}^{n}(v_i : T_i), \forall_{i=1}^{p}(h_i : \phi_i), \forall(h_a : \hat{A}), \psi$$

The ground-truth $\bar{A}$ satisfies

$$\forall_{i=1}^{n}(v_i : T_i), \forall_{i=1}^{p}(h_i : \phi_i), (\psi \leftrightarrow \bar{A})$$

Therefore, we have

$$\forall_{i=1}^{n}(v_i : T_i), \forall_{i=1}^{p}(h_i : \phi_i), \forall(h_a : \hat{A}), \bar{A}$$

i.e.,

$$\forall_{i=1}^{n}(v_i : T_i), \forall_{i=1}^{p}(h_i : \phi_i), \hat{A} \to \bar{A}$$

$\square$

Notably, for find-unique-one problems (only one valid answer exists) with queriable $a$, if it is expressed in D-FPS as in Appendix E.2 and concludes a direct answer of the form $\hat{A} = (\boxed{a} = \hat{a})$, a backward proof is not required anymore (completeness implies soundness). Because of the uniqueness of $\bar{a}$ and Assumption 3.4, the valid answer term $\hat{a}$ must be the unique one.

## F. More Analysis of RPE

We evaluate RPE on 300 examples from the test set of xVerify (Chen et al., 2025). Each sample is manually annotated with a correctness label between an informal solution and a ground-truth answer. We use xFinder (Yu et al., 2025) to extract informal answers from the informal solutions and use DeepSeek-V3 (DeepSeek-AI et al., 2024) with 4-shot demonstrations to transform the informal answers and corresponding ground-truths into RPE statements. (See Appendix H.7 for detailed prompt)

### F.1. Ablation of Restricted Tactics

We ablate the restricted tactic set in RPE by adding tactics one by one. As shown in Table 4, the current tactic set keeps $100\%$ precision and achieves the best recall and Cohen's kappa.

### F.2. Error Analysis on xVerify Benchmark

On the xVerify benchmark consisting of 300 examples, RPE results in 0 false positives and 4 false negatives. Their error types are categorized as follows:

- Failure of xFinder (2 false-negatives). xFinder incorrectly extracts an intermediate result as the answer.
- Intolerance of numerical error (1 false-negative). RPE is based on formal verification; thus, imperfect floating-point approximation should not be passed. In this false-negative, the answer $0.4667$ and ground-truth $\frac{7}{15}$ are not equivalent under RPE.
- Insufficiency of proof automation (1 false-negative). RPE incorrectly determines propositions $x^2/3 + y^2/4 - 1 = 0$ and $\frac{x^2}{3} + \frac{y^2}{4} = 1$ as inequivalent. This type of error does not significantly affect the following proposed benchmarks since most of the answers are simple terms instead of complicated propositions.

*Table 5.* Preference scores of FPS and D-FPS methods.

| Comparison | Preference Model | Mean | Min | Max |
|---|---|---|---|---|
| Hybrid CoT - Proof Search | GPM | 0.77 | 0.28 | 1.00 |
|  | RLHFlow | 0.80 | 0.27 | 0.98 |
| Hybrid CoT - Whole-Proof Generation | GPM | 0.76 | 0.05 | 1.00 |
|  | RLHFlow | 0.81 | 0.13 | 0.97 |
| Whole-Proof Generation - Proof Search | GPM | 0.01 | -0.36 | 0.57 |
|  | RLHFlow | 0.47 | 0.07 | 0.97 |

### F.3. Rejected Case Study in Experiments

Here, we analyze answers rejected by RPE from In-Context Learning and Hybrid CoT:

- $a = \sqrt{180}/2 \leftrightarrow a = 3 * \sqrt{5}$. This is correctly rejected due to insufficient simplification;
- $a = \{x \mid x^2 - 5x + 6 > 0\} \leftrightarrow a = \{x \mid x < 2 \vee x > 3\}$. This is correctly rejected since the answer is directly constructed from the original problem and is insufficiently solved;
- $a = \sum_{i,j,k}\left(\frac{2^{(j+k-2)}}{3^{(i+k-2)}\times 5^{(i+j-2)}} + \frac{2^{(j+k-3)}}{3^{(i+k-3)}\times 5^{(i+j-3)}}\right) \leftrightarrow a = \frac{17}{21}$. This is correctly rejected since the answer is directly constructed from the original problem and is insufficiently solved;
- $2004\%12 = 0 \leftrightarrow a = 0$. This is correctly rejected since the submitted answer is irrelevant to the queriable $a$;
- $a = \{f(\mathbb{R}^2) \mid f \in \mathbb{R}[x,y]\} \leftrightarrow a = \{\{x\} \mid x \in \mathbb{R}\} \cup \{[x,\infty) \mid x \in \mathbb{R}\} \cup \{(-\infty,x] \mid x \in \mathbb{R}\} \cup \{(-\infty,x) \mid x \in \mathbb{R}\} \cup \{(x,\infty) \mid x \in \mathbb{R}\} \cup \{\mathbb{R}\}$. This is correctly rejected since the answer is directly constructed from the original problem and is insufficiently solved.

## G. More Analysis of Experiment Results

### G.1. Preference Experiment

In this experiment, to validate D-FPS's better alignment with human preference, we quantitatively evaluate preference scores on the solutions from FPS and D-FPS using preference models.

We extract formal solutions of commonly solved problems by Proof Search (InternLM2.5-StepProver), Whole-Proof Generation (DeepSeekProver-V1.5), and Hybrid CoT. All informal comments are removed to ensure a fair comparison since InternLM2.5-StepProver seldom generates informal thoughts, while Hybrid CoT always incorporates informal steps for flexible reasoning. Two preference models, general-preference/GPM-Gemma-2-2B (Zhang et al., 2025) and RLHFlow/pair-preference-model-LLaMA3-8B (Dong et al., 2024), are evaluated with prompt templates in Appendix H.8.

Results are shown in Table 5, where Hybrid CoT consistently outperforms Proof Search and Whole-Proof Generation by a clear margin.

### G.2. Specialist Provers on D-FPS

Specialist LLM provers exhibit severe shortcut behavior on D-FPS. As shown in Table 6, they often typecheck by submitting vacuous answers such as `True`, but their solved rates remain low after RPE evaluation.

## H. Prompt Templates

### H.1. Prompting Generalist LLMs for FPS

```
Given an informal math problem and a corresponding Lean 4 formal statement, please
    construct a formal proof.
Please assume the following header code has already been executed, and do not add any
    imports or openings.
The 'problem ... find ... s.t. ... : ... := solve' is a syntactic sugar that creates two
    goals: 'Answer' and 'Proof', representing the answer to the problem and a proof of
```

*Table 6.* Specialist LLM provers on D-FPS. `True` reports the percentage of vacuous `True` submissions.

| Benchmark | Model | Typecheck% | Solved% | True% |
|---|---|---|---|---|
| FormalMath500 | DeepSeek-Prover-V1.5-RL | 59.17% | 5.17% | 42.11% |
| | Goedel-Prover-V2-8B | 100.00% | 5.94% | 97.95% |
| MiniF2F-Solving | DeepSeek-Prover-V1.5-RL | 46.40% | 3.47% | 48.45% |
| | Goedel-Prover-V2-8B | 100.00% | 9.87% | 97.63% |
| PutnamBench-Solving | DeepSeek-Prover-V1.5-RL | 62.96% | 0.31% | 43.89% |
| | Goedel-Prover-V2-8B | 98.77% | 0.00% | 99.03% |

```
    the answer's correctness.
```lean4
import Mathlib
import Aesop
import FormalProblemSolving.Basic
```

Here is one example:

```lean4
/-
## Problem
Given three real numbers a, b, c with a > 0, find the minimizer x₀ of ax² + bx + c over ℝ.
-/

problem Q_quad_min (a b c : ℝ)
find (x₀ : ℝ) s.t.
  (ha : a > 0)
  (f : ℝ → ℝ)
  (hf : f = fun x => a * x^2 + b * x + c)
: (∀ x, f x ≥ f x₀) := solve
```

# Formal Proof
```lean4
intro x
let vertex := - b / (2 * a)
have h_factorization : (a * x^2 + b * x + c) - (a * vertex^2 + b * vertex + c) = a * (x -
    vertex)^2 := by
    have ha_ne_zero : 2 * a ≠ 0 := by linarith
    dsimp [vertex]
    field_simp [ha_ne_zero]
    ring
rw [ge_iff_le, ← sub_nonneg]; subst hf
case Answer := - b / (2 * a)
change 0 ≤ (a * x^2 + b * x + c) - (a * vertex^2 + b * vertex + c)
rw [h_factorization]; apply mul_nonneg
· linarith
· apply sq_nonneg
```

Now, please generate a formal proof for the following problem.
Please DO NOT call any tool.
Please answer instantly and only output the formal proof (do not output `problem ... find
    ... s.t. ... : ... := solve`).

```lean4
{FPS_PROBLEM}
```
```

## H.2. Prompting Generalist LLMs for FTP

```
Given an informal math problem and a corresponding Lean 4 formal statement, please
    construct a formal proof.
Please assume the following header code has already been executed, and do not add any
    imports or openings.
```lean4
import Mathlib
import Aesop
```

Here is one example:

```lean4
/-
## Problem
Given three real numbers a, b, c with a > 0, find the minimizer x₀ of ax² + bx + c over ℝ.
-/

example (a b c : ℝ)
(x₀ : ℝ) ({H_SUBMISSION_NAME}: x₀ = (- b / (2 * a)))
  (ha : a > 0)
  (f : ℝ → ℝ)
  (hf : f = fun x => a * x^2 + b * x + c)
: (∀ x, f x ≥ f x₀)
:= by
```

# Formal Proof
```lean4
intro x
have h_factorization : (a * x^2 + b * x + c) - (a * x₀^2 + b * x₀ + c) = a * (x - x₀)^2 :=
    by
    have ha_ne_zero : 2 * a ≠ 0 := by linarith
    simp [{H_SUBMISSION_NAME}]
    field_simp [ha_ne_zero]
    ring
rw [ge_iff_le, ← sub_nonneg]; subst hf
change 0 ≤ (a * x^2 + b * x + c) - (a * x₀^2 + b * x₀ + c)
rw [h_factorization]; apply mul_nonneg
· linarith
· apply sq_nonneg
```

Now, please generate a formal proof for the following problem.
Please DO NOT call any tool.
Please answer instantly and only output the formal proof (do not output `example ... :=
    by`).

```lean4
{FPS_PROBLEM}
```
```

## H.3. Proof Search

The following uses InternLM2.5-StepProver ([Wu et al., 2025](#)) as an example.

We directly use the Lean goal state as the `STATE` input.

```
---
NAME: {THEOREM_FULL_NAME}
```

```
---
PROOF_BEFORE: {PROOF_BEFORE}
---
STATE_BEFORE: {STATE}
---
TACTIC:
```

## H.4. Whole-Proof Generation

The following uses DeepSeek-Prover-V1.5 (Xin et al., 2024b) as an example.

Since existing models are all trained for theorem proving, a special proxy formal statement and initial tactic sequence are used in the prompt template to avoid out-of-distribution inputs. The statement and the tactic sequence work identically to the `problem` and `solve` DSL. They create the same subgoals, `Answer` and `Proof`.

For thinking models such as Goedel-Prover-V2 (Lin et al., 2026), we also use the proxy statement and tactic sequence. During generation, we insert them upon detecting the start of a Lean statement.

```lean4
Complete the following Lean 4 code with explanatory comments preceding each line of code:

```lean4
import Mathlib
import Aesop
set_option maxHeartbeats 0
set_option maxRecDepth 100000
set_option tactic.hygienic false
set_option pp.fullNames true
set_option pp.funBinderTypes true
set_option pp.piBinderTypes true

/-- {INFORMAL_PROBLEM}-/

example :
∀(v_1 : T_1)⋯(v_n : T_n)(h_1 : φ_1)⋯(h_p : φ_p),
∃(a : T_a), ψ_1 ∧ ⋯ ∧ ψ_q := by
    intros v_1⋯v_n  h_1⋯h_p
    apply Exists.intro ?Answer ?Proof
```

The `example` statement:

$$\forall(v_1 : T_1)\cdots(v_n : T_n)(h_1 : \phi_1)\cdots(h_p : \phi_p),$$
$$\exists(a : T_a), \psi_1 \wedge \cdots \wedge \psi_q := \text{by}$$

## H.5. Direct ICL

```
Given an informal math problem and a corresponding Lean 4 goal state, please construct a
    formal proof deducing the answer.
Please assume the following header code has already been executed, and do not add any
    imports or openings.
```lean4
import Mathlib
import Aesop
```

Here are some examples:

# Informal Problem
"""
{INFORMAL_PROBLEM}_i
"""
# Goal State
```lean4
{FORWARD_STATE}_i
```
# Formal Proof
```lean4
{FORWARD_SOLUTION}_i
```
```

```
---

 (10-shot demonstrations)...

---

Now, please generate a formal proof for the following problem.

# Informal Problem
"""
{INFORMAL_PROBLEM}
"""
# Goal State
```lean4
{FORWARD_STATE}
```
```

## H.6. Hybrid CoT

```
Given an informal math problem and a corresponding Lean 4 goal state, please think step
    by step and construct a formal proof deducing the answer.
Please assume the following header code has already been executed, and do not add any
    imports or openings.
```lean4
import Mathlib
import Aesop
```

Here are some examples:

# Informal Problem
"""
{INFORMAL_PROBLEM}_i
"""
# Goal State
```lean4
{FORWARD_STATE}_i
```
# Formal Proof
```lean4
{FORWARD_SOLUTION_WITH_INFORMAL_THOUGHTS}_i
```

---

 (10-shot demonstrations)...

---

Now, please generate a formal proof for the following problem.

# Informal Problem
"""
{INFORMAL_PROBLEM}
"""
# Goal State
```lean4
{FORWARD_STATE}
```
```

## H.7. RPE Benchmark Formalization

```
Given two answers to some problem, please construct a Lean 4 statement to assert their
    equivalence. Don't try to prove them; only formalize the statement. The following are
    some examples:

# Answer A
\( x < -\frac{4}{3} \) or \( x > 0 \)
# Answer B
$(-\infty ,-\frac{4}{3})\cup (0,+\infty )$
# Formal Statement
'''lean4
example : {x : ℝ | x < (-4/3 : ℝ) ∨ x > (0 : ℝ)} = (Set.Iio (-4/3 : ℝ)) ∪ (Set.Ioi (0 : ℝ
    )) := by sorry
'''

# Answer A
\frac{1 + \sqrt{1 + 8n}}{2}
# Answer B
(1 + (1 + 8n)^(1/2)) / 2
# Formal Statement
'''lean4
example (n : ℝ) : (1 + Real.sqrt (1 + 8 * n)) / 2 = (1 + (1 + 8 * n)^(1/2 : ℝ)) / 2 :=
    by sorry
'''

# Answer A
(m−1)(n−1)
# Answer B
$\binom{m+n-2}{m-1}$
# Formal Statement
'''lean4
example (m n : ℕ) : (m-1) * (n-1) = Nat.choose (m+n-2) (m-1) := by sorry
'''

# Answer A
364000
# Answer B
$3.64 \times 10^5$
# Formal Statement
'''lean4
example : (364000 : ℚ) = (3.64 : ℚ) * (10^5) := by sorry
'''

Now, please process the two answers:
# Answer A
{answer_a}
# Answer B
{answer_b}
```

## H.8. Preference Model

### User Message

```
The following is a problem and its Lean 4 formalization.

# Informal Problem
"""
{informal_problem}
"""

# Formal Problem
'''lean4
{formal_problem}
'''
```

*Table 7.* Environment options.

| Option | Value |
|--------|-------|
| `maxHeartbeats` | 0 |
| `maxRecDepth` | 100000 |
| `tactic.hygienic` | false |
| `pp.fullNames` | true |
| `pp.funBinderTypes` | true |
| `pp.piBinderTypes` | true |

```
Please use Lean 4 code to solve the following problem step by step.
```

**Assistant Message**

```lean4
{FORMAL_SOLUTION_WO_INFORMAL_STEPS}
```

## I. Detailed Experiment Settings

### I.1. Hyperparameters

**Proof Search.** Following (Wu et al., 2025), we set the number of children nodes in each expansion $S = 32$, the search budget $K = 600$, the sampling temperature $T = 0.7$, and the maximum token limit for each tactic as 256. The detailed prompt template can be found in Appendix H.3. The adopted Lean 4 interface, Pantograph (Aniva et al., 2024), requires tactic applications to be specific to goals. Therefore, in each expansion step, we evenly allocate $S$ tactic applications to each goal.

**Whole-Proof Generation.** Following the setting of DeepSeek-Prover-V1.5 (Xin et al., 2024b), we set the search budget $K = 128$, sampling temperature $T = 1.0$, top-p value 0.95, and the maximum token limit of 2048 by default. For model-specific evaluation, the token budget follows each model's context window: Goedel-Prover-V2 (Lin et al., 2026) and DeepSeek-Prover-V2 (Ren et al., 2025) use an adaptive budget up to about 8192 tokens, Kimina-Prover (Wang et al., 2025) uses $T = 0.6$, top-p 0.95, and an adaptive budget up to about 8096 tokens, and STP (Dong & Ma, 2025) uses a maximum token limit of 1024. The detailed prompt template is in Appendix H.4.

**Generalist LLM Prompting and DSP+.** For FPS prompting baselines, we use $K = 16$ for DeepSeek-V3.2 (DeepSeek-AI et al., 2025b) and Sonnet-4.5, with sampling temperature $T = 1.0$, top-p value 0.95, and maximum token limit 8192. For DSP+, the sketch-generation budget is $K = 8$, with $T = 1.0$, top-p 0.95, and maximum token limit 8192; the BFS-Prover component uses a tree-size budget of 16, beam width 4, tactic temperature 1.1, and maximum tactic length 64 tokens.

**Direct & Hybrid CoT.** For D-FPS, we use $K = 16$ for DeepSeek-V3.2 (DeepSeek-AI et al., 2025b) and Sonnet-4.5, with sampling temperature $T = 0.7$. The maximum token limit is 8192. The 10-shot demonstrations are randomly sampled from the MATH (Hendrycks et al., 2021) train set, with 2 for each subject. The detailed prompt templates are in Appendix H.5 and Appendix H.6. We only run forward reasoning to disentangle forward solving from backward proving and directly evaluate the resulting answers with RPE.

### I.2. Lean Environment Settings

All relevant open-source projects are summarized in Table 8 for reproducibility. Special thanks to the authors of these excellent projects :)

The options of Lean 4 environments are summarized in Table 7. In each Pantograph (Aniva et al., 2024) REPL, we import `Mathlib` and `Aesop`, while the opened namespaces vary according to problems.

*Table 8.* Open-source projects used in this paper.

| Name | Github Link | Version |
|---|---|---|
| **Lean 4** (Moura & Ullrich, 2021) | https://github.com/leanprover/lean4 | v4.15.0 |
| **Mathlib 4** (mathlib Community, 2020) | https://github.com/leanprover-community/mathlib4 | v4.15.0 |
| **Aesop** (Limperg & From, 2023) | https://github.com/leanprover-community/aesop | v4.15.0 |
| **Pantograph** (Aniva et al., 2024) | https://github.com/lenianiva/Pantograph | v0.2.25 |

## J. Limitations and Future Work

**Extension to Find-One Problems and Open Conjectures.** Our current benchmarks and the RPE metric focus on problems with closed-form or canonical explicit solution representations, including complete solution-set characterizations for *find-all* problems. In contrast, generic *find-one* problems (identify a single valid instance among many candidates), e.g., counter-example construction and open conjecture exploration, remain underexplored. These problems present unique evaluation challenges as they lack a single canonical ground truth. Future work may construct dedicated benchmarks for find-one problems. We believe the exploration of prompt-based evaluation models (e.g., LLM-as-a-Judge with detailed rubrics) can help capture human alignment.

**Data Scarcity and Autoformalization.** As a novel domain, FPS and D-FPS are currently constrained by data scarcity. A scalable solution lies in *autoformalization*, i.e., automatically translating informal problems and solutions into formal language. However, ensuring the semantic fidelity of autoformalized statements remains an open challenge. Future research may focus on autoformalization pipelines to scale FPS to broader mathematical domains and effectively bridge the gap between natural language reasoning and formal verification.

**Subject Coverage and Formalization Depth.** In constructing **FormalMath500**, we excluded `Geometry` and `Counting & Probability`. In these specific domains, the primary difficulty often stems from the formalization process itself (e.g., translating diagrams or combinatorial scenarios into logic) rather than the subsequent derivation. Once fully formalized, these problems often collapse into trivial arithmetic computations, losing their original reasoning depth. Future work may explore reasoning-preserving formalizations that retain the cognitive complexity of the original informal problems.

