# OpenReview forum: "Beyond Theorem Proving: Formulation, Framework and Benchmark for Formal Problem-Solving"
_ICML.cc/2026/Conference — ICML 2026 spotlight_

### Official Review · Reviewer_NBg3 · 2026-03-11

**Soundness:** 3
**Presentation:** 2
**Significance:** 2
**Originality:** 3
**Overall Recommendation:** 3
**Confidence:** 3

**Summary:**

This paper focuses on formalizing the process of solving mathematical problems. The authors propose two new tasks, Formal Problem-Solving and Deductive Formal Problem-Solving, and provide formal descriptions for both. They demonstrate how these problem-solving processes can be obtained within a formal environment through metaprogramming. To support these tasks, the authors construct three datasets, FormalMath500, MiniF2F-Solving, and PutnamBench-Solving, derived from existing formal mathematics benchmarks. Experiments with off-the-shelf theorem-proving models show that the solving task is more challenging than the proving task and reveal the current capabilities of these models on the proposed tasks.

**Compliance With Llm Reviewing Policy:**

Affirmed.

**Final Justification:**

From my point of view, the main limitation of this work lies in its significance. The proposed FPS/D-FPS frameworks are primarily evaluated on exercise-style problems (where one can synthesize much more data from the original sources), which already fit well into this formulation, making it unclear how much new capability they provide beyond existing approaches. It remains uncertain whether the framework generalizes to more complex or realistic mathematical reasoning tasks. Although the rebuttal offers clarification, it does not fully resolve these concerns, and the noticeable performance drop from miniF2F (more exercise-style) to PutnamBench (less exercise-style) further highlights this limitation.

If the AC considers the significance concern to be less critical, I would not object to the paper being accepted.

**Key Questions For Authors:**

1. In Definition 3.8 (2), the term human intuition appears to be somewhat ambiguous. Since human intuition can be incorrect, biased, or inconsistent across individuals, its use in a formal definition may introduce subjectivity. Could the authors clarify how human intuition should be interpreted in this context and how it is operationalized within the formal framework?

2. In Section 4.1, the paper mentions a manual refinement step during the dataset construction process. Could the authors provide more details on the extent of human involvement, such as the approximate percentage of instances that required manual refinement? For a hybrid data construction pipeline, quantifying the degree of automation versus manual intervention would help readers better understand the scalability and reproducibility of the dataset creation process.

**Limitations:**

Yes

**Strengths And Weaknesses:**

**Strengths**:

1. The theoretical formulation of Formal Problem-Solving (FPS), Deductive Formal Problem-Solving (D-FPS), and Restricted Propositional Equivalence (RPE) in Section 3 is novel and technically sound. It establishes a formal framework that supports the rigor of the proposed design. In particular, RPE, which is supported by human evaluation, could be useful as an evaluation metric for other research in formal mathematics.

2. The paper constructs three benchmarks, FormalMath500, MiniF2F-Solving, and PutnamBench-Solving, based on existing formal mathematical datasets. The data construction process is clearly described. These benchmarks could be valuable resources for future research on formal mathematical reasoning.

**Weaknesses**

1. The proposed framework may have limited generalizability across broader areas of mathematics. The experiments mainly focus on relatively elementary problems that are easier to formalize. It remains unclear how well the proposed approach would perform on other types of mathematical reasoning tasks, such as proofs involving universal claims or more complex mathematical structures.

2. The presentation of FPS and D-FPS is somewhat confusing. They appear to be introduced as both new tasks and frameworks for transforming mathematical problems into alternative formal formulations, rather than as methods that directly improve theorem proving. However, based on the results in Tables 1–3, FPS and D-FPS seem to function more like benchmarks, where models are evaluated on their ability to perform these tasks. Clarifying whether FPS/D-FPS are intended primarily as evaluation tasks, datasets, or methodological frameworks would improve the paper’s clarity.

---

> ### Author Rebuttal · Authors · 2026-03-31
>
> We greatly appreciate your constructive comments. We sincerely hope the following paper **updates (U)** and **responses (R)** align with your suggestions.
>
> - U1. Polish the expressions of FPS/D-FPS
> - U2. Incorporate discussions in R4 into a new Appendix
>
> > W1. Is FPS/D-FPS generalizable to other types of mathematical reasoning tasks, e.g., proofs involving universal claims or more complex mathematical structures?
>
> **R1.** Thank you for this comment, it sparked a heated discussion among the authors. In short, FPS / D-FPS can formalize any mathematical problems which can be expressed as a proof-of-existence statement (e.g., "find $x$ s.t. $P(x)$"  → $\exists x. P(x)$ where $P$ may contain other free variables) in Lean 4. Definition 3.2 provides a flexible formalization of problems that can contain free variables, hypotheses about them, the queried variable, and the constraints of the queried variable.
>
> For example, see the problem in `FormalProblemSolving/PutnamBenchSolving/47.lean` of the anonymous repo (line 108 right). For all `(f : ℝ → ℝ), (g : ℝ → ℝ)`, the answer depends on `f` and `g` and returns another function.
>
> > W2. Presentation of FPS and D-FPS
>
> **R2.** Sorry for the confusion in our presentation. You are right that FPS/D-FPS are both evaluation tasks and methodological frameworks for formalizing mathematical problems. Our three constructed benchmarks have parallel formalizations in both FPS/D-FPS. Therefore, FPS/D-FPS in Table 1-3 represents which framework the problems are formalized in.
>
> We will polish the presentation in Table 1~3 by replacing words like "FPS benchmarks" with "benchmarks formalized in FPS".
>
> > Q1. How human intuition should be interpreted in this context and how it is operationalized within the formal framework?
>
> **R3.** Human intuition here represents how human judges whether an answer is semantically correct (logical correctness is included in Definition 3.8 (1) "$P(\hat a)$ holds"). Human prefer simplified answer $-\frac{b}{2a}$ to unsimplified answer $-\frac{2b}{4a}$; concrete answer $0$ to tautological answer $\lim_{n\rightarrow \infty}\frac{1}{n}$.
>
> We acknowledge the term may introduce perceived subjectivity. Operationally, it is grounded in RPE's comparison against ground-truth answers curated by domain experts, ensuring alignment and consistency.
>
> > Q2. More details on the extent of human involvement during the dataset construction process.
>
> **R4.** Good question! TLDR: 5% ~ 47% data require manual refinement. It depends on the data source (formalized from informal problem or adapted from formal statement). Due to the space limit, please refer to [R1,R2] to Reviewer RmZR for more details.
>
> ---
>
> Due to the space limit, we place some **more responses (M)** here.
>
> **M4.**
>
> Table 1 shows that in Proven%, Kimina-Prover is about 10% higher than Goedel-Prover on both FormalMath500 and MiniF2F-Solving, while in Solved%, the gaps are about 20% respectively.
>
> This difference can be interpreted from two perspectives.
> 1. Starting point. Goedel-Prover has relatively strong theorem-proving capabilities, while its problem-solving capabilities are weaker. Therefore, improving 10% in Proven% is as significant as 20% in Solved%.
> 2. Data source. We indirectly analyze their training data distributions by comparing their formalizers with the informal problems in FormalMath500. The formalization modes of problems "find $x$ s.t. $P(x)$" with answer $a$ are categorized into:
>  - Exist + GT: $(\exists x. P(x)) \wedge P(a)$
>  - GT: only $P(a)$
> The "Exist & GT" resembles the FPS formalization more.
>
> |Model|Exist+GT(%)|GT(%)|Other(%)|
> |-|-|-|-|
> |Goedel-V1|1.29%|91.21%|7.49%|
> |Kimina|7.24%|88.63%|4.13%|
> |Goedel-V2|31.27%|67.70%|1.03%|
>
> The percentage of problems formalized under "Exist & GT" shows a strong positive correlation with the problem-solving performance (Solved%) of the three models.
>
> **M5.**
> Backward goals are not separately selected/formulated but are automatically updated from the forward stage (upon an answer is derived).
>
> After loading the D-FPS problem, Lean returns one proof state with three sub-goals:
> - `Answer`: A proposition answering the problem.
> - `Forward`: Asserting that the `Answer` is a necessary condition of the ground-truth (completeness). The target (content after `⊢`) is placeholdered with `?Answer`.
> - `Backward`: Asserting that the `Answer` is a sufficient condition of the ground-truth (soundness). The `_h_answer` is a placeholder with `?Answer`.
> The goals `Forward` and `Backward` are coupled with the goal `Answer`. Please refer to Appendix D for detailed examples.
>
> In the example of Fig. 2, after the formal answer `x₀ = -b / (2 * a)` is derived, it is automatically filled into the `Backward` goal.
>
> It is possible that agents derive a tautological answer, e.g.,`∀ (x : ℝ), f x ≥ f x₀`. RPE in the subsequent evaluation will detect and reject these.

---

> > ### Author Rebuttal · Reviewer_NBg3 · 2026-03-31
> >
> > Thank you to the authors for their rebuttal. I do not have further questions.
> >
> > My decision to maintain my scores is primarily based on concerns about the significance of the work. As far as I understand, most excercise-style samples in benchmarks such as miniF2F can be represented in the described logical format. However, for more complex benchmarks, such as ProofNet, many problems do not naturally fit into this format.
> >
> > From my perspective, performance on excercise-style samples is already relatively strong across existing approaches. Therefore, greater emphasis should be placed on evaluating methods on problems that are closer to realistic mathematical practice, rather than primarily focusing on exercise-style benchmarks.
> >
> > For these reasons, I will maintain my original assessment of the paper.

---

> > > ### Author Response · Authors · 2026-04-06
> > >
> > > Thank you for this thoughtful follow-up (F) comment. We appreciate the opportunity to sharpen the framing and presentation of this paper :)
> > >
> > > > F1. most excercise-style samples in benchmarks such as miniF2F can be represented in the described logical format. However, for more complex benchmarks, such as ProofNet, many problems do not naturally fit into this format.
> > >
> > > **R5.1.** To clarify, we respectfully disagree with the premise that the setting we study is merely "exercise-style". Our focus is constructive solving of unknown objects rather than proving known propositions. This includes difficult undergrad-level problems in PutnamBench-Solving and also naturally connects to broader answer-construction settings, e.g., open problems.
> > >
> > > **R5.2.** Formal theorem proving benchmarks like ProofNet focus on proving known propositions and are therefore complementary rather than direct substitutes for our setting.
> > >
> > > **R5.3.** In fact, our current benchmarks already include complex undergraduate-level problems: PutnamBench-Solving contains 324 problems from undergraduate-level competitions spanning Algebra, Analysis, Number Theory, Geometry, Linear Algebra, Combinatorics, Abstract Algebra, Probability, and Set Theory.
> > >
> > >
> > > > F2. performance on excercise-style samples is already relatively strong across existing approaches
> > >
> > > **R6.** If the "existing approaches" refer to informal reasoning, we agree that current methods can often derive plausible answers without end-to-end formal verification. However, this is different from the formally verified problem-solving regime studied here. In our setting, performance under FPS / D-FPS is still far from strong. On the PutnamBench-Solving benchmark, the best solved rate is only 1.23% (Table 1 and [M1] to Reviewer RmZR).
> > >
> > > > F3. greater emphasis should be placed on evaluating methods on problems that are closer to realistic mathematical practice
> > >
> > > **R7.** Excellent comment! Many open problems in mathematics are in the form of "Find $x$ s.t. $P(x)$". They can be formalized and explored under the FPS / D-FPS frameworks. We totally agree with this suggestion and are actively extending to open questions and conjectures.
> > >
> > > Thank you again for your constructive reviews. We will update the camera-ready by:
> > > - **U3.** Add clearer discussions to Sec. 1, Paragraph 2 to avoid potential misunderstandings;
> > > - **U4.** Integrate the discussion in [R7] into Appendix J "Extension to Find-One Problems", and rename the subsection to "Extension to Find-One Problems and Open Conjectures".

---

### Official Review · Reviewer_FMhC · 2026-03-12

**Soundness:** 3
**Presentation:** 3
**Significance:** 3
**Originality:** 2
**Overall Recommendation:** 4
**Confidence:** 4

**Summary:**

This paper analyses the difference between problem-solving and proving. Problem-solving involved generating an answer to the problem whereas as proving is given the answer and is expected to generate and proof that reaches the answer. The paper evaluates this issue across three different datasets where they ensure that they refactor the problem to suit the problem-solving vs proving aspect.

**Compliance With Llm Reviewing Policy:**

Affirmed.

**Final Justification:**

Thank you for the effort you've put into the rebuttal, I've improve my overall score to weak accept now.

**Key Questions For Authors:**

1) line 099 - 100 second column, there's a floating 's'

2) Can you explain your claim from line 368-370 column as to why the gap between Solved% and Typecheck% would be from tautological answers or unsimplified terms? Did you conduct an evaluation of this?

3) Why was DeepSeek-V3 not used in Table 1? And why is this chosen for D-FPS experiments?

**Limitations:**

yes

**Strengths And Weaknesses:**

Strengths:
1) An important problem has been raised with the paper i.e. the difficulty of generating an answer, which specifically target a drawback of current proving ability assumptions.
2) The problem is presented with theoretical framework and an example.
3) The experiments are conducted on a range of datasets including a large amount of questions, and multiple models which better support the findings.

Weaknesses:
1) The related work section omits work that are directly related to this paper e.g. PutnamBench's "without_solution" setting, Conjecturing: An Overlooked Step in Formal Mathematical Reasoning, Enumerate-Conjecture-Prove: Formally Solving Answer-Construction Problems in Math Competitions. Moreover, RPE is very similar to RETHINKING AND IMPROVING AUTOFORMALIZATION: TOWARDS A FAITHFUL METRIC AND A DEPENDENCY RETRIEVAL-BASED APPROACH but this is not mentioned.
2) The type of questions considered seem to only be for problem statements that have a specific closed answer, i.e. -b/2a. How do your experiment support the problem where the problem statement requires a proof of existence (line 269) that is not explicit? If this is the case, then you work on a particular subset that seem to already have other means of being tested for soundness and completeness given that the answer exists in the statement.
3) I understand that the D-FPS setting is a subset within the refactored benchmark that you created. What is its size and distribution? And how did you select and formulate backward direction questions? For instance in the example you give, why don't you generate the formal answer to be f(x_0) \ge f(x), why is this given?

---

> ### Author Rebuttal · Authors · 2026-03-31
>
> We greatly appreciate your constructive comments and invaluable review! Briefly, we have **updated (U)** the following points in camera-ready:
>
> - U1. Discuss more related works in Sec. 2.
> - U2. Add comparison between RPE and BEq into Appendix B. "Evaluation of Answer Correctness".
> - U3. Incorporate response R3 into Sec. 3.2
> - U4. Fix typo in line 099.
> - U5. Add evaluation of generalist LLMs on FPS to Sec. 5.
>
> Our detailed **responses (R)** are as follows.
>
> > W1.1 More related works.
>
> **R1.** Thank you for mentioning these great works :) We will add explicit citations of these works to Sec. 2.
>
> Moreover, line 127 right "post-hoc verification" and Fig. 1 (b) are exactly the pattern that the above literature follows: separating the construction of answers (aka. conjectures) and formal proofs. It only validates the answer, leaving the chain-of-thought answer derivation unverified.
>
> > W1.2 Difference between RPE and BEq.
>
> **R2.** Excellent question! In fact, that paper is cited at line 644 left.
>
> RPE and BEq (_Rethinking and improving autoformalization: towards a faithful metric and a Dependency Retrieval-based approach_) fundamentally differ in 3 aspects:
> 1. Task focus. RPE evaluates the equivalence between math answers and targets at FPS/D-FPS, while BEq evaluates the equivalence between formal statements and targets at autoformalization.
> 2. Different proving target. RPE tries to prove propositional equality, while BEq tries to prove that two statements imply each other.
> 3. Different implementation. RPE is symbolic, while BEq combines with an LLM.
>
> > W2.1 The type of questions considered seems to only be for problem statements that have a specific closed answer, i.e., b/2a.
>
> **R3.** Thank you for pointing out this ambiguity! FPS can formalize beyond problems with a specific closed-form answer.
>
> It can formalize both _find-one_ (find an arbitrary valid answer) and _find-all_ (find all valid answers) problems. Only the former may have one unique closed-form answer. A find-all problem can be reinterpreted as "find one answer set, while the set covers all valid answers", which is a special form of find-one problem.
>
> The sentence at line 273 (left) may cause misunderstanding. We will polish it in camera-ready.
>
> > W2.2 How does your experiment support the problem where the problem statement requires a proof of existence that is not explicit?
>
> **R4.** To clarify, proof-of-existence statements like "Prove that $\exists a. P(a)$" are not our primary objective. We focus on problem-solving tasks, which require both deriving an explicit answer and proving its correctness. Proof-of-existence that does not require an explicit answer more resembles traditional FTP (formal theorem proving). If the Proof-of-Existence problem requires an explicit answer, it can be formalized into FPS.
>
> > W3.1 What is D-FPS's coverage of the benchmark
>
> **R5.** The D-FPS framework covers all problems in the three benchmarks. Theorem 3.6 shows D-FPS is at least as expressive as FPS w.r.t. find-all problems. Moreover, Appendix E.2 provides an adaptation from a find-all problem's FPS formalization into a semantically equivalent D-FPS formalization.
>
> > W3.2 how did you select and formulate backward direction questions? For instance, in the example you give, why don't you generate the formal answer to be f(x_0) \ge f(x), why is this given?
>
> **R6.** Backward goals are not separately selected/formulated but are automatically updated from the forward stage (upon an answer is derived). Please refer to [M5] to Reviewer NBg3 for detailed response.
>
> > Q1. Typo in line 099-100
>
> **R7.** Thank you, we will fix in camera-ready :)
>
> > Q2. Explaining the difference between Solved% and Typecheck%
>
> **R8.** We have manually examined the rejected answers of "Typecheck but not Solved" problems, and find 98.7% of them are true negatives. Among true negatives, 10.3% are unsimplified terms and 89.7% are tautological answers.
>
> > Q3.1 Why was DeepSeek-V3 not used in Table 1?
>
> **R9.** Because the FPS framework resembles traditional FTP (formal theorem proving), and we empirically find general-purpose LLMs underperform in FTP tasks.
>
> We have evaluated `DeepSeek-V3.2` and `Claude-Sonnet-4.5` and the detailed results are in [M1] to Reviewer RmZR. Results show specialist LLM provers outperform general-purpose LLMs.
>
> > Q3.2 Why is this chosen for D-FPS experiments
>
> We answer from two aspects:
> 1. Why not choose specialist LLM provers for D-FPS experiments?
>
> **R10.1.** Preliminary experiments show specialist LLM provers suffer from shortcut behavior on D-FPS. During writing, this failure was omitted to avoid distracting readers from the big picture.
>
> Due to the space limit, please refer to [M2] to Reviewer RmZR for detailed statistics.
>
> 2. Why choose DeepSeek-V3 as the general-purpose LLM?
>
> **R10.2.** Because DeepSeek-V3 offered the best cost-performance tradeoff at the time of our experiments. Please refer to [M3] to Reviewer RmZR for more experiments.

---

> > ### Author Rebuttal · Reviewer_FMhC · 2026-04-04
> >
> > Thank you for the work you've put into responding to my queries. Can you please make sure that you explicitly state that these are for solving questions that are closed-form in your revised version?

---

> > > ### Author Response · Authors · 2026-04-06
> > >
> > > Sure. We will update (U) the camera-ready by:
> > > - **U6.1.** Polish Sec. 3.4, clarify that, in the current implementation, find-all problems are evaluated through a canonical closed-form answer object representing the complete solution set;
> > > - **U6.2.** Polish the beginning of Sec. 4, explicitly state that the current benchmarks focus on solving questions with closed-form answers;
> > > - **U6.3.** Polish Appendix J, clarify that the study focuses on problems with a unique or canonical closed-form answer.
> > >
> > > Thank you again for this thoughtful follow-up suggestion, which really helped improve the clarity of this paper :)

---

### Official Review · Reviewer_hS2N · 2026-03-13

**Soundness:** 3
**Presentation:** 4
**Significance:** 2
**Originality:** 3
**Overall Recommendation:** 5
**Confidence:** 3

**Summary:**

This paper proposes to extend formal theorem proving into a formal problem-solving framework that requires finding an answer and verifying its correctness for discovering unknown answers.
The paper thus introduces two frameworks: (1) FPS (Formal Problem-Solving) that unifies derivation and verification by constructing a pair of answer term and its proof within the formal environment, (2) and D-FPS that decouples formal problem solving into a forward derivation stage and a backward verification stage. The paper then introduces the RPE (Restricted Propositional Equivalence) metric for semantic equivalence between answer terms using restricted proof automation. The paper also proposes a data annotation pipeline and constructs three benchmarks with informal-formal parallel problems and answers: FormalMath500, MiniF2F-Solving, and PutnamBench-Solving. Multiple whole-proof generation methods and proof search methods are evaluated on the FPS and D-FPS benchmarks.

**Compliance With Llm Reviewing Policy:**

Affirmed.

**Final Justification:**

The authors have addressed my concerns. So I have raised my score to 5.

**Key Questions For Authors:**

* Please justify the selection of tactics for restricted tactic set T. And how does the variation of T affect the validation effectiveness?
* The performance on PutnamBench-Solving is significantly lower than on FormalMath-500 and MiniF2F-Solving. Could you explain the reasons for this gap and demonstrate model performance with some comparative examples?
* Goedel-Prover and Kimina-Prover show similar performance in terms of Proven%, but differ significantly in Typecheck% and Solved%. What might be the reasons behind this?

**Limitations:**

Yes.

**Strengths And Weaknesses:**

Strengths:
* The proposed FPS and D-FPS avoid the cheating behavior in formal theorem proving, where implicit assignment leads to tautology or an incorrect term to close the proof.
* The introduced metric RPE helps evaluate human-aligned problem solving, which achieves high precision, recall, and alignment.
* This work is supported by rigorous proofs.
* The paper includes detailed case studies.
* The paper is clearly written, well-organized, and easy to follow.


Weaknesses:
* The evaluation on FPS can be more comprehensive. For example, include more recent  Goedel-Prover-V2 [1] and provers with different training paradigms, e.g., STP [2], DSP+ [3], and general-purpose LLMs, e.g., DeepSeek-V3.2 [4], Sonnet-4.5 [5].
* It needs further explanation for the implementation of D-FPS using DeepSeek-V3 instead of LLM provers.
* The evaluation on D-FPS is restricted to a single backbone, which leads to an incomplete understanding of the task and the performance of frontier models.
* It is unclear how D-FPS error modes in Table 3 were obtained and what the definition of each error mode is.
* Benchmarks construction requires supplementing with a human evaluation to demonstrate data quality and alignment with human preferences.
* The FPS and D-FPS benchmark results in Tables 1 and 2 should include human performance.

[1] Lin, Yong, Shange Tang, Bohan Lyu, et al. “Goedel-Prover-V2: Scaling Formal Theorem Proving with Scaffolded Data Synthesis and Self-Correction.” arXiv:2508.03613. Preprint, arXiv, August 5, 2025. ~[https://doi.org/10.48550/arXiv.2508.03613](https://doi.org/10.48550/arXiv.2508.03613)~.
[2] Dong, Kefan, and Tengyu Ma. “STP: Self-Play LLM Theorem Provers with Iterative Conjecturing and Proving.” February 11, 2025. ~[https://doi.org/10.48550/arXiv.2502.00212](https://doi.org/10.48550/arXiv.2502.00212)~.
[3] Cao, Chenrui, Liangcheng Song, Zenan Li, et al. “[DSP+] Reviving DSP for Advanced Theorem Proving in the Era of Reasoning Models.” *NeurIPS 2025*, June 13, 2025. ~[https://doi.org/10.48550/arXiv.2506.11487](https://doi.org/10.48550/arXiv.2506.11487)~.
[4] DeepSeek-AI, Aixin Liu, Bei Feng, et al. “DeepSeek-V3 Technical Report.” arXiv:2412.19437. Preprint, arXiv, February 18, 2025. ~[https://doi.org/10.48550/arXiv.2412.19437](https://doi.org/10.48550/arXiv.2412.19437)~.
[5] Anthropic. System Card: Claude Sonnet 4.5. September 29, 2025. ~https://www.anthropic.com/news/claude-sonnet-4-5~

---

> ### Author Rebuttal · Authors · 2026-03-31
>
> Thank you for your appreciation of this paper and your insightful review! Due to the review policy, we could not update the PDF submission. Our **updates (U)** to the paper are as follows:
>
> - U1. Add more evaluation on FPS into Sec. 5.
> - U2. Add evaluation of whole-proof generation methods on D-FPS into the Appendix.
> - U3. Add more evaluation on D-FPS into Sec. 5.
> - U4. Add ablations of RPE tactics into the Appendix.
>
> Below are our detailed **responses (R)** to your invaluable suggestions and comments :)
>
> > W1. More comprehensive evaluation on FPS.
>
> **R1.** Sure. Experiments reveal that specialist LLM provers outperform general-purpose LLMs and DSP+, and `Goedel-Prover-V2-8B` shows the best performance. Due to the space limit, please refer to [M1] to Reviewer RmZR for details.
>
> > W2. Further explanation for the implementation of D-FPS using DeepSeek-V3 instead of LLM provers.
>
> **R2.** Good question! Honestly, we have a long exploration about this. Preliminary experiments show specialist LLM provers suffer from shortcut behavior on D-FPS. During writing, this failure was omitted to avoid distracting readers from the big picture.
>
> Due to the space limit, please refer to [M2] to Reviewer RmZR for detailed statistics.
>
> > W3. More evaluation on D-FPS with multiple backbones
>
> **R3.** Sure. The method Hybrid CoT significantly outperforms Direct ICL and the model `DeepSeek-V3.2` outperforms `Sonnet-4.5`. Due to the space limit, please refer to [M3] to Reviewer RmZR for detailed statistics.
>
> > W4. Explain the definitions of D-FPS error modes.
>
> **R4.** Thank you for this suggestion! We will add brief explanations of each error mode to Sec. 5 in the camera-ready version. Their definitions are as follows:
> - Length: Incomplete responses truncated due to token limits.
> - Rejection: Request rejected by API providers.
> - Format: Response unparsable due to invalid format.
> - Submission: The submitted answer is irrelevant to the queriable.
> - Answer: The submitted answer is wrong (determined via HuggingFace Math-Verify).
> - Solution: The formal solution fails the Lean 4 check.
>
> These numbers are obtained by analyzing the experiment results of DeepSeek-V3 in the D-FPS framework.
>
> > W5. Human evaluation to demonstrate data quality and alignment with human preferences.
>
> **R5.** Yes. All formalizations are reviewed by human experts to ensure quality and human-alignment. please refer to [R1,R2] to Reviewer RmZR for more details.
>
> > W6. Human performance of FPS and D-FPS benchmarks in Tables 1 & 2.
>
> **R6.** Thank you for this suggestion! This is an interesting experiment, and we too have wondered about it.
>
> However, we cannot obtain precedents of such experiments from existing formal benchmarks, including MiniF2F, ProofNet and PutnamBench. Conducting rigorous human studies requires careful protocol design and appropriate subject recruitment, which exceeds the rebuttal timeline.
>
>  We will pursue this in camera-ready if time & budget permit :)
>
> > Q1. Justification of the selection of tactics for restricted tactic set T. How does the variation of T affect the validation effectiveness?
>
> **R7.** Inspired by human preferences that answers should be simple, neat, and concise, we decide to use tactics of canonicalizers, normalizers, definitional equivalence, and simple searchers to implement this intuition.
>
> We have ablated the tactics in RPE one by one. The results are as follows:
>
> |Tactic|Precision|Recall|Cohen's Kappa|
> |-|-|-|-|
> |Only`rw_search`|100.00%|78.17%|0.7904|
> |+`rfl`|100.00%|81.69%|0.8245|
> |+`norm_num`|100.00%|95.77%|0.9598|
> |+`aesop`|100.00%|96.48%|0.9665|
> |+`ring_nf`(Current)|100.00%|97.18%|0.9732|
> Composing RPE tactics one-by-one continuously improves the recall and Cohen's Kappa to human preference. And the current implementation of RPE performs the best.
>
> We will incorporate this ablation into the Appendix :)
>
> > Q2. Explain the performance gap between PutnamBench-Solving and other benchmarks.
>
> **R8.** Because their difficulties and domains are different. PutnamBench-Solving is reconstructed from PutnamBench, which contains problems in an undergraduate-level math competition, the William Lowell Putnam Mathematical Competition. And it covers broad subjects. However, FormalMath500 and MiniF2F-Solving concentrate on high-school competition level problems and focus on narrower domains (detailed in Sec. 4.2). Previous literature on formal theorem proving also demonstrates a significant gap between MiniF2F and PutnamBench, such as Goedel-Prover-V2 (92.2% vs. 86/644≈13.4%).
>
> Due to the space limit, please refer to our anonymous repo (line 107 right) for comparative examples :)
>
> > Q3. Comparison between Goedel-Prover and Kimina-Prover.
>
> **R9.** Due to the space limit, please refer to [M4] to Reviewer NBg3 for detailed statistics.

---

> > ### Author Rebuttal · Reviewer_hS2N · 2026-04-03
> >
> > Thanks for the detailed response. I have raised the score to 5.

---

### Official Review · Reviewer_RmZR · 2026-03-13

**Soundness:** 3
**Presentation:** 3
**Significance:** 3
**Originality:** 3
**Overall Recommendation:** 5
**Confidence:** 4

**Summary:**

The authors present a formal problem solving (FPS) and Deductive FPS framework for Lean, a semantic answer equivalence metric called PRE and three formal benchmarks derived from existing benchmarks to evaluate FPS. The experiments suggest that current systems perform worse on FPS tasks than proving tasks, suggesting that FPS is a harder and under-explored challenge in this space.

**Compliance With Llm Reviewing Policy:**

Affirmed.

**Key Questions For Authors:**

Can you comment on the dataset generation process? How much is manually refined vs how much passes phase II on its first try?

**Limitations:**

Yes

**Strengths And Weaknesses:**

1) Presentation Clarity: The structure of the paper is very clear. Figures 1 and 2 in particular are very helpful for understanding FPS and Deductive FPS. The paper has a very robust related works section.

2) Significance and Novelty: The notion that solving is harder than proving is an interesting idea and deserves a lot more thought in this space than it has been given. The results, that solving is harder than proving are interesting, although I wonder how much of this can be attributed to the fact that proving is the dominant task for reasoning models that use Lean. The benchmark itself is significant.

3) The new metric of Restricted Propositional Equivalence for evaluation is strong and significantly boosts my confidence in the soundness of the work. The alignment with Cohen’s
Kappa also helps boost confidence.

Weaknesses:

1) I am slightly concerned about transparency in the dataset generation process. The authors mention a manual refinement process, but it is not clear how much of the dataset is manually refined vs how much passes phase II on its first try.

---

> ### Author Rebuttal · Authors · 2026-03-31
>
> Thank you for your helpful suggestions! Our detailed **responses (R)** are as follows :)
>
> > Q1.1. Can you comment on the dataset generation process?
>
> **R1.** Yes. Thank you for pointing out this ambiguity. In fact, Phase III manually reviews and revises all formalizations. We will reorganize Sec. 4.1 in the camera-ready version: Move "If a sample fails in Phase II, human experts manually correct the logical specification" to Phase II; Rename "Automated Verification" into "Logic Verification" and "Manual Refinement" into "Manual Review". Add "two human experts review and revise all formalizations to ensure semantic fidelity" to Phase III.
>
> > W1. & Q1.2. How much is manually refined vs how much passes phase II on its first try?
>
> **R2.** 100% problems are manually reviewed to ensure semantic correctness, ~90% problems pass phase II on its first try.
>
> In Phase I, the initial formalization of problems is generated by rejective sampling with 4-shot prompting DeepSeek-V3-1226. The coarse accuracy is 39.27%.
>
> Then, in Phase II and Phase III, we systematically review and revise the benchmarks. Below are the detailed statistics:
> - Contradictory: All contradictory (ill-posed) problems that do not pass Phase II are repaired by human experts.
> - Sound: All problems whose soundness proof fails to construct (unsure whether to be sound) are specially marked, and human experts in Phase III pay more attention to check them.
> - Accepted: Problems that are correct.
> - Revised: Problems that require human revision.
>
> |Benchmark|Automated Verification||Human Review|||
> |-|-|-|-|-|-|
> ||Contradictory|Sound|Reviewed|Accepted|Revised|
> |FormalMath500|7.24%|63.31%|100.00%|53.23%|46.77%|
> |MiniF2F-Solving|3.47%|69.60%|100.00%|77.07%|22.93%|
> |Putnam-Solving|0.00%|1.85%|100.00%|95.06%|4.94%|
>
> We have **updated (U)** the paper the following points and the main text is now 9 pages long. Due to the review policy, we were unable to update the PDF submission.
>
> - U1. Polish Sec. 4.1. _Construction Pipeline_.
> - U2. Add statistics of the construction pipeline in Sec. 4.1.
>
> ---
>
> Due to the space limit, we place some **more results (M)** here.
>
> **M1.** More comprehensive evaluation on FPS.
>
> Whole-proof generation (WPG) methods (`Goedel-Prover-V2-8B`, `STP_model_Lean`), general-purpose LLMs (prompting `DeepSeek-V3.2` and `Sonnet-4.5` with the demonstration in Fig. 2), and DSP+ (using `DeepSeek-V3.2`+`BFS-Prover-V1-7B`, combining their official prompt and the demonstration in Fig. 2) in the FPS framework are evaluated.
> |Benchmark|Method|Model|%Proven|%Typecheck|%Solved|
> |-|-|-|-|-|-|
> |FormalMath500|WPG|STP_model_Lean|51.94%|9.30%|6.46%|
> |||Goedel-Prover-V2-8B|80.10%|68.48%|58.91%|
> ||Prompt|DeepSeek-V3.2|39.79%|18.35%|14.99%|
> |||Sonnet-4.5|30.23%|7.75%|6.72%|
> ||DSP+|DeepSeek-V3.2+BFS-Prover-V1-7B|20.93%|30.49%|28.42%|
> |MiniF2F-Solving|WPG|STP_model_Lean|60.80%|1.33%|1.07%|
> |||Goedel-Prover-V2-8B|86.13%|69.60%|66.40%|
> ||Prompt|DeepSeek-V3.2|42.40%|17.87%|16.27%|
> |||Sonnet-4.5|33.33%|8.53%|8.27%|
> ||DSP+|DeepSeek-V3.2+BFS-Prover-V1-7B|23.47%|23.20%|21.60%|
> |PutnamBench-Solving|WPG|STP_model_Lean|0.62%|10.80%|0.00%|
> |||Goedel-Prover-V2-8B|1.85%|10.49%|1.23%|
> ||Prompt|DeepSeek-V3.2|0.93%|2.47%|0.00%|
> |||Sonnet-4.5|0.62%|0.31%|0.00%|
> ||DSP+|DeepSeek-V3.2+BFS-Prover-V1-7B|0.31%|9.57%|0.00%|
>
> **M2.** Specialist LLM Prover on D-FPS.
>
> Specialist LLM provers show severe shortcut behavior on D-FPS: the model tends to use `True` as the answer regardless of the problem. We have tried several different prompting styles, and all failed. We hypothesize this may be because LLM provers are fine-tuned on certain tasks, and their instruction-following capabilities are degraded.
>
> |Benchmark|Model|%Typechecked|%Solved|
> |-|-|-|-|
> |FormalMath500|DeepSeek-Prover-V1.5-RL|59.17%|5.17%|
> ||Goedel-Prover-V2-8B|100.00%|5.94%|
> |MiniF2F-Solving|DeepSeek-Prover-V1.5-RL|46.40%|3.47%|
> ||Goedel-Prover-V2-8B|100.00%|9.87%|
> |PutnamBench-Solving|DeepSeek-Prover-V1.5-RL|62.96%|0.31%|
> ||Goedel-Prover-V2-8B|98.77%|0.00%|
>
> The percentage that models vacuously submit `True` as answers is as follows:
> `DeepSeek-Prover-V1.5-RL`
> - FormalMath500: 42.11%
> - MiniF2F-Solving: 48.45%
> - PutnamBench-Solving: 43.89%
>
> `Goedel-Prover-V2-8B `
> - FormalMath500: 97.95%
> - MiniF2F-Solving: 97.63%
> - PutnamBench-Solving: 99.03%
>
> **M3.** More evaluation on D-FPS.
>
> Direct ICL and Hybrid CoT with `DeepSeek-V3.2` and `Sonnet-4.5` on D-FPS are evaluated.
> |Benchmark|Method|Model|%Typechecked|%Solved|
> |-|-|-|-|-|
> |FormalMath500|DirectICL|DeepSeek-V3.2|30.49%|29.46%|
> |||Sonnet-4.5|20.67%|20.16%|
> ||HybridCoT|DeepSeek-V3.2|50.65%|48.58%|
> |||Sonnet-4.5|24.03%|23.26%|
> |MiniF2F-Solving|DirectICL|DeepSeek-V3.2|41.60%|40.80%|
> |||Sonnet-4.5|28.53%|28.00%|
> ||HybridCoT|DeepSeek-V3.2|58.93%|57.33%|
> |||Sonnet-4.5|29.33%|28.53%|
> |PutnamBench-Solving|DirectICL|DeepSeek-V3.2|0.93%|0.00%|
> |||Sonnet-4.5|0.31%|0.00%|
> ||HybridCoT|DeepSeek-V3.2|7.41%|1.54%|
> |||Sonnet-4.5|0.00%|0.00%|

---

> > ### Author Rebuttal · Reviewer_RmZR · 2026-04-04
> >
> > Thank you for the response, it adequately addresses my question. I maintain my score of accept.

---

### Decision · Program_Chairs · 2026-04-30

**Decision:**

Accept (spotlight)

**Comment:**

This paper introduces Formal Problem-Solving (FPS), a framework encompassing the end-to-end problem-solving process in Lean 4 and D-FPS (Deductive FPS), which enforces a rigorous chain-of-thought structure, aligning formal derivation with human reasoning steps. To support this direction, the paper constructs three benchmarks via the manual refactoring of over 1,000 problems: FormalMath500, MiniF2F-Solving, and PutnamBench-Solving, and proposes RPE (Restricted Propositional Equivalence), a symbolic metric that evaluates semantic correctness beyond brittle string matching. The reviewers like the novelty and the technical contributions of this work. However, please pay attention to the suggestions on the empirical results suggested by the reviewers to improve your final version.